# Simultaneous measurement of excitation-contraction coupling parameters identifies mechanisms underlying contractile responses of hiPSC-derived cardiomyocytes

Berend J. van Meer [1], Ana Krotenberg[1], Luca Sala[1,6], Richard P. Davis [1], Thomas Eschenhagen [2,3], Chris Denning[4], Leon G.J. Tertoolen[1] & Christine L. Mummery[1,5]*

Cardiomyocytes from human induced pluripotent stem cells (hiPSC-CMs) are increasingly recognized as valuable for determining the effects of drugs on ion channels but they do not always accurately predict contractile responses of the human heart. This is in part attributable to their immaturity but the sensitivity of measurement tools may also be limiting. Measuring action potential, calcium flux or contraction individually misses critical information that is captured when interrogating the complete excitation-contraction coupling cascade simultaneously. Here, we develop an hypothesis-based statistical algorithm that identifies mechanisms of action. We design and build a high-speed optical system to measure action potential, cytosolic calcium and contraction simultaneously using fluorescent sensors. These measurements are automatically processed, quantified and then assessed by the algorithm. Multiplexing these three critical physical features of hiPSC-CMs allows identification of all major drug classes affecting contractility with detection sensitivities higher than individual measurement of action potential, cytosolic calcium or contraction.

[1] Dept. of Anatomy and Embryology, Leiden University Medical Center, Einthovenweg 20, 2333 ZD Leiden, The Netherlands. [2] Dept. of Experimental Pharmacology and Toxicology, University Medical Center Hamburg Eppendorf, Martinistraße 52, 20246 Hamburg, Germany. [3] DZHK (German Center for Cardiovascular Research), partner site Hamburg/Kiel/Lübeck, Hamburg, Germany. [4] Dept. of Stem Cell Biology, Centre for Biomolecular Sciences, University of Nottingham, University Park, Nottingham NG7 2RD, UK. [5] Dept. of Applied Stem Cell Technologies, University of Twente, Drienerlolaan 5, 7522 NB Enschede, The Netherlands. [6] Present address: Istituto Auxologico Italiano, IRCCS, Center for Cardiac Arrhythmias of Genetic Origin, Laboratory of Cardiovascular Genetics, Via Zucchi 18, 20095 Cusano Milanino, MI, Italy. *email: c.l.mummery@lumc.nl

Human induced pluripotent stem cell derived cardiomyocytes (hiPSC-CM) are increasingly used to study mechanisms underlying drug responses and disease in vitro, with action- or field potential measurements by microelectrode arrays translating well to clinically relevant arrhythmia or cardiotoxicity classification[1–4]. Pharma and regulatory authorities are thus already implementing such approaches in early drug discovery and cardiotoxicology assessment. However, analytical methods that relate these electrophysiological parameters to contraction and cytosolic calcium transients, also salient features of cardiomyocyte physiology, are still limited. This makes clinically relevant interpretation of drug responses affecting cardiomyocyte contractility challenging[5].

Under normal conditions, contraction is the result of excitation-contraction coupling (EC coupling). This is a complex cascade of ion channel activities, internal calcium storage and contractile filament availability with the action potential (AP), cytosolic calcium flux (Ca) and the actual contraction (Co) being the most important physical transients[6]. The mechanisms of action (MOAs) through which cardioactive drugs affect contraction typically target ion channels or -pumps, adenosine-, muscarinic- and adrenergic receptors, phosphodiesterases (PDE), myosin or elements of thin filaments such as troponin C. The effect of some of these cardioactive drugs would not be expected in hiPSC-CM cultures (for example, antagonists of adrenergic and adenosine receptors require the presence of respective agonists in the culture medium) while often the effect of other drugs would be expected but is undetectable because of hiPSC-CM immaturity[7]. Improving hiPSC-CM maturity, either by altering cellular configuration (e.g. 2D or 3D and cell alignment, which is highly correlated with maturity)[8–10] or refined measurement methods[11] are factors reported to overcome the lack of response in such cases, but are still limited in providing mechanistic insight into hiPSC-CM pharmacological responses. Measurement methods that give insight into AP, Ca and Co simultaneously are needed and they would allow assessment of the complete EC coupling of a single cardiomyocyte beat cycle and thereby reducing data variability and experimental bias compared to asynchronous measurements. We therefore hypothesize that simultaneous quantification of the changes of AP, Ca and Co in response to drugs will provide insight into the contractile MOA affected, higher detection sensitivity of those mechanisms and allow proper contractile risk classification of drugs.

We thus report the development of tools to gain the necessary mechanistic insight into hiPSC-CM physiology. First, building on literature, we develop a hypothesis-based statistical analysis algorithm to test the likelihood of occurrence for each of the most important mechanisms via which drugs affect cardiomyocyte contractility. Next, we engineer a high-speed system capable of measuring and quantifying AP, Ca and Co simultaneously by means of fluorescent sensors ANNINE-6plus, Rhod-3 and CellMask Deep Red in combination with our recently developed optical contraction software tool MUSCLEMOTION[12]. Using this Triple Transient Measurement (TTM) system we measure and quantify the response of commercial hiPSC-CMs to a set of drugs that covers the complete range of potentially affected contractile mechanisms.

We find that simultaneous TTM in combination with our algorithm allows automatic and unbiased identification of the MOAs affected by the test set of compounds and increased the sensitivity of detecting contractile MOAs compared to separate assessment of AP, Ca and Co.

## Results

**Hypothesis table**. To develop an algorithm capable of identifying contractile MOAs affected in response to drugs, we first inventoried potential effects on the kinetics of cardiomyocyte AP, Ca and Co. A schematic is shown in Fig. 1. Three general possibilities defined the first set of categories: enhanced contraction (Co$^+$), reduced contraction (Co$^-$) and no effect on contraction (Co$^=$). The most important contractile MOAs underlying Co$^+$ in cardiomyocytes are (i) Co$_{Ca}^+$: increased free cytosolic calcium concentration ($[Ca^{2+}]_i$) via for example calcium- and sodium channel agonists[13,14], cardiac glycosides[15] and human ether-a-go-go related gene (hERG) channel blockers[16]; (ii) Co$_{Myosin}^+$: direct thick- or thin myosin filament activation[17] or (iii) Co$_{cAMP}^+$: enhancement of the complete contractile cascade via higher levels of the second messenger 3′,5′-cyclic adenosine monophosphate (cAMP) by beta-adrenergic stimulation[18] or PDE3 inhibition[19].

Main mechanisms resulting in Co$^-$ are (i) Co$_{Ca}^-$: reduction in $[Ca^{2+}]_i$ either via changes in the activity of the L-type calcium channel, sarcoplasmic reticulum calcium ATPase (SERCA) or ryanodine receptors type 2[20], or (ii) Co$_{tox}^-$: via general cardiotoxicity, often associated with mitochondrial dysfunction or more generally with increased mitochondrial Ca$^{2+}$ buffering capacity, apoptosis and necrosis[21]. These Co$^-$ mechanisms do not include contractile quiescence that occurs as a consequence of excessive AP duration (APD) prolongation, since these compounds are often Co$^+$ before failing to repolarize in time to meet the imposed stimulation frequency.

Two general class of drugs have been excluded in the algorithm: beta blockers and myosin inhibitors. Since no beta-adrenergic agonists are present in in vitro cultures with defined medium, beta blockers are not relevant in this study. Regarding the myosin inhibitors, this group is small and the mechanisms are not specific. For example, while blebbistatin reduces contractility, it also affects cardiomyocyte electrophysiology[22].

The expected effects on AP, Ca and Co kinetics per MOA are listed in the hypotheses table (Table 1). Due to the heterogeneity of the drugs within each class, it is often possible for some compounds to have mixed or even diverse effects on the kinetic parameters although being grouped by their main effect on Co. For this reason, we indicated in these cases what the effects are that, from a theoretical point of view, we do not expect for each specific class of compound (i.e. NOT ↓; NOT ↑; NOT =). For example, digoxin and E-4031 are both part of Co$_{Ca}^+$, while they shorten[23] and prolong[24] the action potential, respectively. While most kinetic parameters in Table 1 are simple measurements of parts of the waveform, action potential triangulation is calculated as the difference between APD at 30% of repolarization and APD at 90% of repolarization[25]. Triangulation is instrumental in detecting proarrhythmic effects.

**Experimental design and statistical analysis algorithm**. Next, we developed an algorithm based on standard statistics to test whether a certain drug response is described by the effects summarized in the Hypotheses Table. The experimental design to test the concentration-dependent drug response in hiPSC-CMs is shown in Fig. 2a. To eliminate variation and time-dependent response effects, all concentrations are compared to the vehicle (0.01% DMSO) measurements in the resulting concentration-response plots (Fig. 2b). These vehicle measurements are also the basis of our analysis algorithm since the probability of a kinetic parameter not changing ('=' in Table 1) is equal to the probability of it being part of the vehicle population, described by its probability density function (PDF). To determine the likelihood of a parameter to be increased, decreased or unchanged, the PDF was normalized to the mean and multiplied by 10 to have a range of 0–10. The resulting function was divided in three corresponding spaces (Fig. 2c), which determine the Probability Scores (PS) for

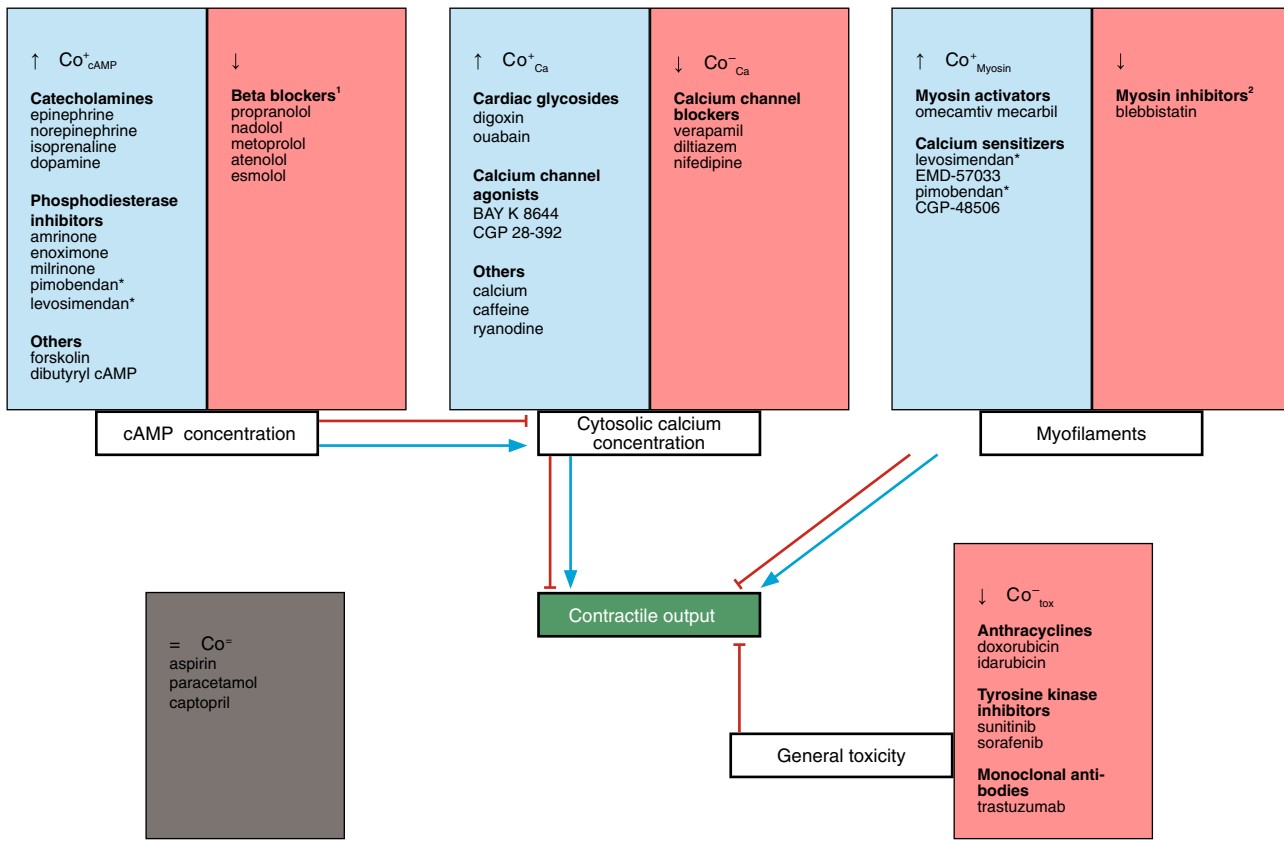

**Fig. 1** Schematic of critical mechanisms affecting cardiomyocyte contraction targeted by drugs. Most therapeutics that target the heart or have off-target effects on the heart fall within the mechanisms shown. 1: Since no beta-adrenergic agonists are present in in vitro cultures with defined medium, this category is not relevant in this study. 2: This group is small and the mechanisms are not specific. For example, while blebbistatin reduces contractility, it also affects cardiomyocyte electrophysiology. Therefore, this category is not included in this study. Asterisk: These compounds exhibit both PDE inhibiting and myofilament $Ca^{2+}$ sensitizing properties, with the dominant mechanism typically depending on the concentration

**Table 1 Hypotheses table**

| | Co= | Co+Ca | Co+Myosin | Co+cAMP | Co−Ca | Co−Tox |
|---|---|---|---|---|---|---|
| **Typical compound** | **Aspirin** | **Ouabain** | **Omecamtiv Mecarbil** | **β-adrenergic agonist** | **Verapamil** | **Doxorubicin (long term)** |
| **References** | 42–44 | 15,45,46 | 17,47,48 | 6,49,50 | 51–53 | 21,54,55 |
| **Kinetic parameters** | | | | | | |
| Action potential (AP) | | | | | | |
| Amplitude | = | = | = | = | ↓ | ↓ |
| $t_{rise}$ | = | ↓ | = | ↓ | NOT ↓ | ↑ |
| $t_{APD}$ | = | NOT = | = | ↓ | ↓ | NOT ↑ |
| Triangulation | = | ↓ | = | ↓ | ↓ | ↑ |
| Calcium transient (Ca) | | | | | | |
| Amplitude | = | ↑ | = | ↑ | ↓ | ↓ |
| $t_{to\ peak}$ | = | NOT ↓ | NOT ↑ | ↓ | ↓ | NOT ↑ |
| $t_{decay}$ | = | NOT ↓ | NOT ↑ | ↓ | ↓ | NOT ↑ |
| Contraction (Co) | | | | | | |
| Amplitude | = | ↑ | ↑ | ↑ | ↓ | ↓ |
| $t_{contraction}$ | = | NOT ↓ | ↑ | ↓ | ↓ | NOT ↑ |
| $t_{relaxation}$ | = | NOT ↓ | NOT ↓ | ↓ | ↓ | NOT ↑ |

Expected effects on kinetic measures of action potential (AP), cytosolic calcium transient (Ca) and contraction (Co) per contractile mechanism of action (MOA). ↓: decrease; ↑: increase; =: no difference. NOT [effect] means all other effects than [effect] might be expected, depending on the specific drug within the MOA category

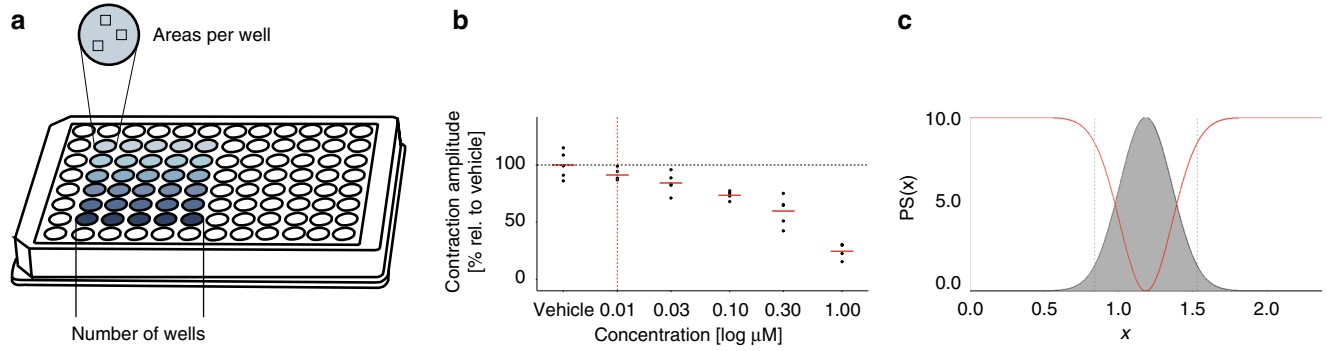

**Fig. 2** Experimental setup and example output. **a** Schematic layout of the experimental setup; **b** output example of contraction amplitude in a concentration-response plot in which individual points show averaged values of a well and the red dotted line indicates the known therapeutic concentration; **c** corresponding probability density function (grey) and $PS_\uparrow$ and $PS_\downarrow$ (red lines) describing the three spaces for the three Probability Scores

each possibility ($\uparrow$, $\downarrow$ or $=$) using the following formulae:

$$PDF(x) = \frac{1}{\sqrt{2\pi \cdot sd_{mean}^2}} \cdot e^{-\frac{(x - mean_{vehicle})^2}{2 \cdot sd_{mean}^2}} \quad (1)$$

$$PS_=(x) = \frac{10 \cdot PDF(x)}{PDF(mean_{vehicle})} \quad (2)$$

$$PS_\uparrow(x) = \begin{cases} 0, & x < mean_{vehicle} \\ 10 - PS_=(x), & x \geq mean_{vehicle} \end{cases} \quad (3)$$

$$PS_\downarrow(x) = \begin{cases} 10 - PS_=(x), & x \leq mean_{vehicle} \\ 0, & x > mean_{vehicle} \end{cases} \quad (4)$$

where $sd_{vehicle}$ is the standard deviation of the vehicle measurements. Since we are interested in finding the most likely hypothesis rather than looking at absolute values resulting from these formulae and $PS_=(x) \gg PS_\uparrow(x)$ and $PS_=(x) \gg PS_\uparrow(x)$ if $\approx mean_{vehicle}$, it is reasonable to assume $PS_\uparrow(x) = 0$ and $PS_\downarrow(x) = 0$ if $x < mean_{vehicle}$ and $x > mean_{vehicle}$, respectively.

The process is repeated for the 10 kinetic parameters according to Table 1 and all PS values are added to reach a total PS between 0 and 100. The hypothesis with the highest total score is most likely to indicate the relevant MOA.

**Development of the Triple Transient Measurement system.** Although a wide range of tools is available to measure the transients separately or as combinations of two, we aimed to measure the three transients AP, Ca and Co simultaneously to decrease the measurement-to-measurement variation and enable multiplexing of all 10 parameters from a single biological sample at exactly the same time. Since most dedicated measurement systems have limited flexibility in the setup itself (e.g. measuring only impedance-based contractility or extracellular electrical field properties by multi electrode arrays; MEAs) or in the throughput or invasiveness (e.g. patch clamp electrophysiology requiring individual cell impalement) an optical system was designed that had a sampling speed >300 frames per second (fps) as required to capture all hiPSC-CM transient kinetics accurately. AP, Ca and Co can be measured optically using fluorescent voltage- and calcium sensitive dyes and fluorescent membrane labelling combined with the analysis algorithm MUSCLEMOTION[12], respectively.

Simultaneous fluorescent imaging can be done using wavelength separation in the emission pathway or by high-speed switching between wavelengths. Since wavelength separation decreases the light intensity and exposure times had to be low due to the sampling speed, high-speed switching was the method of choice in combination with a multiband filter to avoid slow mechanical switching. In turn, this limited the

choice of dyes since they had to have no or minimal overlapping excitation wavelengths. ANNINE-6plus, Rhod 3 and CellMask Deep Red were selected together with the appropriate excitation LEDs emitting at 470 nm, 560 nm and 656 nm (Supplementary Fig. 1). To investigate the extent to which the optical sensors affected baseline electrophysiological responses, hiPSC-CMs previously derived from patients with Jervell and Lange-Nielsen Syndrome (JLNS)[26] were plated on MEAs and the spontaneous electrophysiological response was measured before and after incubation with the dyes (Supplementary Fig. 2). Inferential statistics with ANOVA was performed and no significant effect of the sensors was found. In addition, values found were similar to those previously reported[27]. Finally, the LEDs and camera where synchronized using a computer with data acquisition card and custom software to reach a sampling speed of 1000 fps, resulting in a sample speed of 333 fps per transient, which allowed us to reach an effective synchronisation among transients. Due to the high sampling speed required, the camera exposure time was limited which, in turn, limited the levels of fluorescent signal that could be detected in the camera; we therefore amplified the light using an image intensifier.

Our algorithm uses vehicle measurements to define the baseline population, so a fixed pacing frequency of 1.2 Hz was chosen for all measurements instead of an adaptive frequency. The latter would have required frequency correction for all parameters but the mathematical models for such correction are imperfect for action potentials and unavailable for calcium and contraction. hiPSC-CMs were stimulated using a method developed in-house to produce a block pulse that was chosen to be as short as possible (2–10 ms) and used a voltage as low as possible (10–18 V) while still activating the cells. The settings were kept consistent while measuring under baseline conditions and after drug addition. All wells were measured, and thus stimulated, only twice and no changes were seen in susceptibility of response to the field stimulation during the second measurement. A 50% medium change during drug addition minimized electrolysis effects. To test whether there was any crosstalk between the fluorescent signals of the different dyes, different wells of Pluricyte hiPSC-CMs where exclusively labelled with each of the three dyes and exposed to all three LED wavelengths individually (Supplementary Fig. 3). Assessment of the fluorescence intensity revealed that ANNINE-6plus and Rhod 3 signals were only present in response to 470 and 560 nm wavelengths, respectively. CellMask Deep Red showed a modulating image intensity due to contraction usable for MUSCLEMOTION analysis only at 656 nm. An overview of the setup is shown in Fig. 3a.

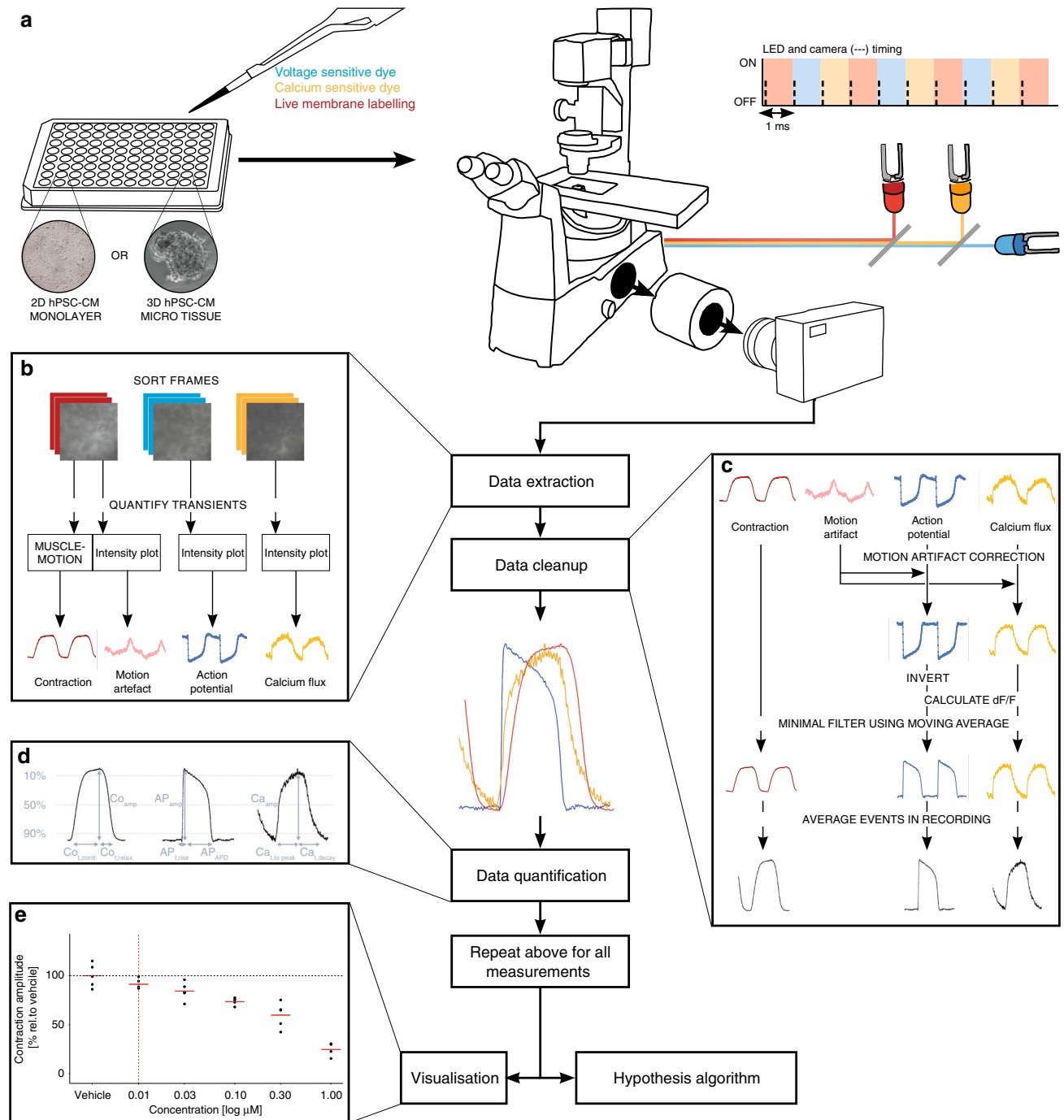

**Fig. 3** Triple Transient Measurement system for determining AP, Ca and Co simultaneously. **a** Hardware setup: the hiPSC-CM model of interest (2D or 3D) is incubated with the voltage sensitive dye (ANNINE-6plus), calcium sensitive dye (Rhod 3) and cell membrane label (CellMask Deep Red) used for measuring contraction. The hiPSC-CMs are excited using three rapidly alternating LEDs which are synchronized with a high-speed camera. The camera receives the emitted signal from the hiPSC-CMs after photon amplification by the image intensifier. The data are then fed into the data processing pipeline. **b** The video frames are split into three stacks that contain the information regarding the AP (blue), Ca (yellow) and Co (red). The information of each stack is extracted by either plotting the intensity changes and/or applying MUSCLEMOTION. **c** The data are then "cleaned" to remove the motion artefacts, invert the signal when necessary, calculate ΔF/F, apply a minimal filter and/or average the events within one recording. **d** The data are quantified by summarizing the kinetic parameters of interest determined in Table 1. **e** Finally, the data are plotted and meanwhile fed into the hypothesis algorithm

**Data extraction and processing**. For each measurement point, 7 s of recording were made at 1000 fps using a spatial resolution of 720 × 720 pixels yielding a file size of 3.6 GB. This large amount of data is then handled using automated data processing algorithms. First, the total file is split into three separate stacks corresponding with the three transients AP, Ca and Co. Next, the image intensity plots are extracted from all three image stacks (Fig. 3b). This yields the raw AP signal, Ca signal and the motion artefact (MA) signal. Since CellMask Deep Red (i.e. the fluorophore that is the origin of the MA signal) is colocalized with the VSD in the membrane of the hPSC-CMs and recorded simultaneously with the VSD signal, we could use the MA signal for

ratiometric correction of the AP and Ca signal[28]. Meanwhile, the Co stack was also analysed using MUSCLEMOTION to quantify hiPSC-CM contraction.

Next, the data were cleaned by applying MA correction, the AP signal inverted (since the signal intensity of ANNINE-6plus decreases with depolarization), the pseudo ratio $\frac{\Delta F}{F_{min}}$ for AP and Ca calculated, a minimally moving average filter (2 frames) applied and finally five events in the recording averaged[29] after checking whether the interpeak distance corresponded to the pacing frequency (Fig. 3c). The data were then summarized by the kinetic parameters described in Table 1 (Fig. 3d). After these steps were repeated for all the measurements, concentration-response plots were generated per kinetic parameter for visual interpretation of the results (Fig. 3e). Finally, the data were fed into the hypothesis algorithm.

**hiPSC-CM drug response**. To validate the setup and algorithm, a set of 12 drug responses was selected that comprised at least all

MOA identified in Table 1 once. Briefly, commercial hiPSC-CM monolayers were cultured in glass-bottomed 96-well plates. Per drug, five concentrations and a vehicle (DMSO) were tested. These conditions were each tested in five different wells and three individual areas were measured per well (Fig. 2a). The area was measured before and after drug- or vehicle incubation (30 min at 37 °C). Representative transients normalized to baseline of vehicle measurements and an illustrative concentration per drug are shown in Fig. 4. The vast amount of data (±600 GB per drug) was then automatically evaluated and quantified.

Next, the analysis algorithm was applied to the data. For each drug at and each concentration, the hypothesis table was populated using the calculations of the probability scores and summarized (Supplementary Table 1). As described earlier, the highest summarized value identified the MOA. The resulting assignments per concentration are shown in Supplementary Table 2 (column 'ALL'). The algorithm accurately indicated the correct MOAs for 10 of the 12 drugs when analysing the response of AP, Ca and Co. A comparison with conventional statistical methods per parameter

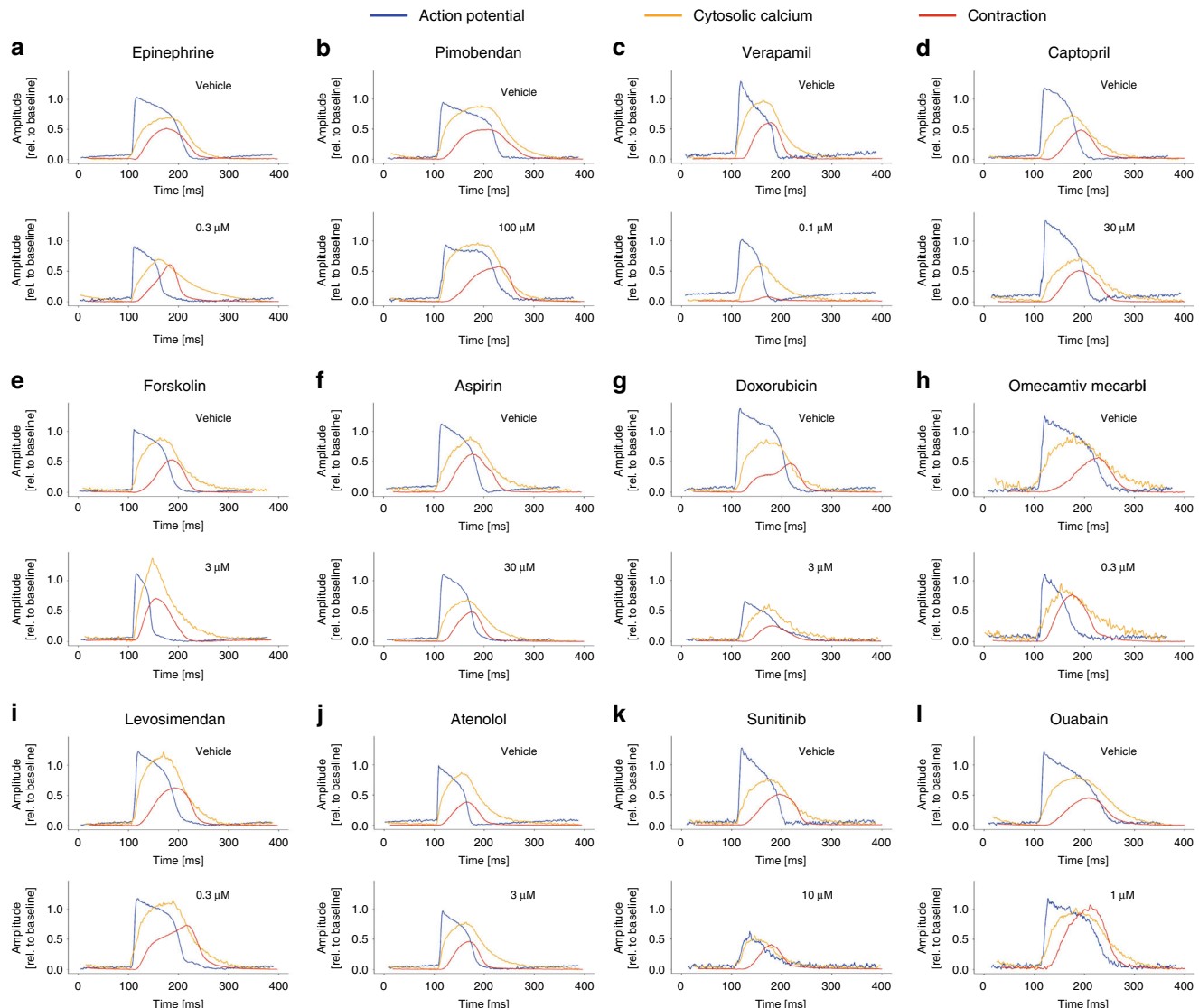

**Fig. 4** Averaged representative transients of hiPSC-CMs in response to vehicle (DMSO) and drugs. All measurements were normalized to their individual baseline measurements. Cells were paced at 1.2 Hz. Simultaneous measurements show action potential (blue), cytosolic calcium (orange) and contraction (red) of Pluricytes under baseline conditions and in response to 0.3 µM epinephrine (**a**), 100 µM pimobendan (**b**), 0.1 µM verapamil (**c**), 30 µM captopril (**d**), 3 µM forskolin (**e**), 30 µM aspirin (**f**), 3 µM doxorubicin (**g**), 0.3 µM omecamtiv mecarbil (**h**), 0.3 µM levosimendan (**i**), 3 µM atenolol (**j**), 10 µM sunitinib (**k**) and 1 µM ouabain (**l**)

per drug per dose can be found in Supplementary Table 3. It is evident that the algorithm is typically not triggered by small changes that might not be physiologically relevant (e.g. captopril at 1 μM) while it does indicate a relevant change in certain cases although no statistical differences are evident in the individual parameters (e.g. omecamtiv mecarbil at 0.03 and 0.3 μM). In some cases, small 'significant' changes in single parameters might not be physiologically meaningful while a combined system of changes does more robustly indicate a relevant cell response.

Both PDE3-inhibiting drugs levosimendan and pimobendan had no effect assigned by the algorithm, whilst the opposite would be expected based on the literature, where the calcium sensitizing- and PDE3-inhibiting effects of the drugs should cause a positive inotropic effect[30,31]. Reviewing the data manually also did not show any changes compared to vehicle. This lack of effect is most likely due to by hiPSC-CM immaturity[32,33] or incubation time. Furthermore, one unexpected observation was made with atenolol, which binds to the β1-adrenoceptor and competes with endogenous neurostimulators. Because neurostimulators are not present in serum-free hiPSC-CM cultures, no effect was expected, but a myosin-related increase in Co was seen at 0.3 μM. However, calcium current enhancement and inhibition have been reported previously in agonist-free conditions[34] which might explain the increase in $AP_{t,rise}$ at 0.3 μM. Therefore, while the assignment of a myosin related contractile increase is incorrect, the assignment of 'no effect' is also questionable. Since the dataset generated is too large to include in standard figures, an interactive online tool was developed to give readers access to all data and both qualitative and quantitative insight in the data and analysis. In addition, the tool allows comparison to traditional statistics. Baseline statistics and raw concentration-response curves are also available to demonstrate the high inter-measurement variation and time-dependent effects, respectively, which is the reason for normalization to the vehicle. The tool is available at https://bjvanmeer.shinyapps.io/TTM-algorithm/.

**Comparison to AP, Ca or Co only**. In addition, the assignments that resulted from only analysing Vo, Ca or Co are listed in Supplementary Table 2 (column 'AP-', 'Ca-' and 'Co only'). As expected, assignments of the individual parameters are not as accurate as analysis of all parameters together and this gives further strength to our combined analysis strategy. The individual parameters are often incorrect in identifying the MOA and more frequently show inconsistent responses throughout the different concentrations. Another issue, which can also be seen in Table 1, is that some parameters are unable to distinguish between certain MOAs (i.e. AP only is unable to distinguish between $Co^=$ and $Co^+_{Myosin}$; Co is unable to distinguish between $Co^+_{Myosin}$ and $Co^+_{Ca}$).

Other platforms that use multi parameter assessments in hPSC-CMs exist but differ from the TTM system. The system developed by Klimas et al. relies on optogenetics and therefore requires cell lines in which the optogenetic actuators have been transfected or stably inserted, adding significant complexity and decreasing flexibility[35]. Furthermore while the channelrhodopsin2 enables the cells to be paced, it has to be controlled by light pulses which limits the time for fluorescence recording and therefore only allows simultaneous measurements of two parameters (voltage and calcium) rather than all three. While the CellOPTIQ system[36] (Clyde Biosciences) can record all three parameters, this can only be performed sequentially and not simultaneously like the TTM system. Additionally, the CellOPTIQ system has a lower frame rate which might limit the accuracy especially for the action potential. However, the resulting output is similar to that obtained with the TTM system, meaning that similar analysis using the hypothesis-based model for the classification of pharmacological effects might be possible. Similarly, Christoph et al. recently published a more complex method to perform accurate 3D voltage mapping[37]. While only suitable for 3D tissues and more focused on the spatiotemporal patterns during fibrillation, the output data from this system might be suitable for the hypothesis-based algorithm.

Automated evaluation of the output data generated by different systems using the hypothesis-based algorithm would enable comparable interpretation of the results. Since the algorithm is based on the vehicle population of the system it would not be limited by differences between absolute values or percentages of change which often make head to head comparison of different techniques difficult or impossible. More generally, user bias is greatly reduced using automated interpretation. However, currently all systems still rely on non-automated manual response interpretation.

In summary, it is now widely recognized that predicting clinical impact of drugs using in vitro and/or in silico models and obtaining more insight in the mechanisms underlying individual responses to drugs or disease mutations is crucial for developing new therapeutic compounds, treatments and strategies. The work here contributes to this by combining multiple functional readouts in one system with automatic quantitative assessment not subject to operator bias. We focussed specifically on contractile mechanisms in cardiac models while others have integrated qualification of other drug types such as chemotherapeutics[38]. Combining such scores might ultimately lead to an integrated cardiac safety score. The majority of hiPSC-CMs remain immature in most standard laboratory formats and are therefore associated with a high inter-sample variability. Whilst methods are emerging that can promote maturation[7,8,39,40], advanced multiplexed tools as described here can help identify immature features and suggest possible methods to address this and reduce inter-sample variability. The method was demonstrated here on monolayer cultures of hiPSC-CMs, but is equally applicable to more complex multicellular and 3D formats. Ultimately, multiplexing not only functional readouts but also those from genetics, metabolomics and proteomics could provide truly predictive models for insight into the cellular mechanisms underlying drug response and pathology in the human heart.

## Methods

**TTM system**. An Eclipse Ti microscope (Nikon) was fitted with a 3 high-power LED system (Mightex; 470 nm, 560 nm and 656 nm) including collimators and excitation filters (Semrock; 470 ± 14 nm, 544 ± 12 nm, 650 ± 7.5 nm). The LEDs were controlled using a custom-made current source that was coupled to a 32 channel PCI DAQ Card (National Instruments). Furthermore, an image intensifier (Photonis, the Netherlands), which was coupled to a high-speed camera (Optronis GmbH), was triggered by the DAQ card using a custom Labview (National Instruments) program. The microscope was fitted with a custom-built environmental chamber that allowed measurements to be performed at 37 °C and 5% $CO_2$. All analyses were performed using ImageJ (NIH) and R (R Foundation for Statistical Computing).

**hiPSC-CM cell culture**. Black glass-bottom 96-well plates (Greiner) were coated with 1:100 Matrigel (Sigma–Aldrich) in DMEM F12 (Sigma–Aldrich). Per drug, one vial of commercial hiPSC-CMs (Pluricytes; Ncardia, catalogue number PCK-1.5, kindly provided by Ncardia) was thawed according to manufacturer's instructions in Pluricyte medium (PCM; Ncardia, catalogue number PCK-1.5, kindly provided by Ncardia). To allow dye incubation and measurements in parallel, cells were distributed over two plates in 15–20 wells (40k cells per well in 100 μL). RevitaCell (1:100, ThermoFisher Scientific) was added to the culture medium to improve cell recovery. Empty wells around the wells with hiPSC-CMs were filled with 200uL of $PBS^{-/-}$ (Gibco) to minimize evaporation. Cells were refreshed with 200 μL PCM medium at day 1 and day 4 or 5 after plating. All measurements for each drug were always performed either on day 5 and 6 or day 6 and 7 after plating.

**Dye and drug incubation**. ANNINE 6-plus (Sensitive Farbstoffe GbR), Rhod-3 (Invitrogen) and CellMask Deep Red (Invitrogen) with stock concentrations 0.5 g per L, 10 mM and 5 mg per mL were mixed in 500 μL PCM in ratios of 1:833, 1:833

and 1:100, respectively. 50 μL of the mix was added to each well and incubated for 20 min at 37 °C. Before starting baseline measurements, wells were refreshed with 200 uL per well of PCM and left for 10 min at 37 °C to recover.

After baseline measurements 100 μL medium was removed and 100 μL of drug solution in PCM (200% of the final concentration) was added and the hiPSC-CMs incubated for 30 min at 37 °C. All drugs were reconstituted in DMSO (Sigma–Aldrich) with solutions prepared to ensure a final concentration of 0.1% DMSO in each well. Vehicle incubations were done similarly using PCM plus 0.1% DMSO. All drugs, apart from ouabain (Sigma–Aldrich), were kindly provided by GlaxoSmithKline (United Kingdom).

**MEA Recordings of hiPSC-CMs**. Extracellular field potentials from hiPSC-CMs plated on MEAs were recorded (recording duration 2 min) using a 96-well Multiwell MEA system (Multichannel Systems GmbH) at 37 °C running the Multiwell-Screen software. Beating monolayers of hiPSC-CMs were incubated with fluorescent reporters following the protocol developed for the TTM System, with DMSO concentrations balanced accordingly. After incubating the hiPSC-CMs with the dyes for 20 min, the beating monolayers were washed with warm BPEL and the electrophysiological properties of the monolayers assessed at two different time points after the washout (2–4 min and 13–15 min). Data were exported with Multiwell-Analyzer and visualised through a custom-made R script. The interpretation of QT and RR intervals was done using Clampfit (analysis module from Axon™ pCLAMP™ 11)[41].

**Reporting summary**. Further information on research design is available in the Nature Research Reporting Summary linked to this article.

## Data availability

We have developed an interactive webtool to access the vast amount of data more easily and intuitively. The tool is available at https://bjvanmeer.shinyapps.io/TTM-algorithm/. The source data underlying Supplementary Table 2, Supplementary Table 3 and Supplementary Fig. 2 are provided as a Source Data file. In addition, the source data of all measurements is provided as Source Data file. In addition, the data can be found at the Dryad Digital Repository under the following https://doi.org/10.5061/dryad.67c92vm.

## Code availability

Source codes for the software are available on https://gitlab.com/bjvanmeer/ttm-algorithm/.

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

## Acknowledgements

This work was supported by the National Centre for the Replacement, Refinement and Reduction of Animals in Research [grant number NC/K000225/1] through the NC3Rs CRACK-IT (full proposal code 35911–259146). We gratefully acknowledge support and input from all consortium members: G. Smith, F. Burton, K. McGlynn (Clyde Bioscience & University of Glasgow, United Kingdom); A. Hansen, I. Mannhardt, U. Saleem (EHT Technologies & University Medical Center Hamburg Eppendorf, Germany); R. Passier, M. Ribeiro (Twente University, the Netherlands); P. Katili, N. Mohd Yusof (University of Nottingham); S. Braam, T. de Korte (Ncardia, the Netherlands); A. Bahinsky, P. Clements, S. Turner (GlaxoSmithKline, United Kingdom); C. Vickers, S. Jackson (NC3Rs, United Kingdom). We are particularly grateful to GSK for contributing to defining and providing the test set of compounds used, and to Ncardia for providing hPSC-CMs and medium. We thank Sander van Berloo, Huybert van de Stadt and Bram de Visser for their indispensable craftsmanship and support in developing the TTM system. Finally, we would like to thank Mervyn Mol, Giulia Campostrini and Elisa Giacomelli for critical reviewing of the manuscript and providing hiPSC-CMs for preliminary testing with the TTM system. C.D. was supported by the British Heart Foundation [grant numbers SP/15/9/31605, RG/15/6/31436, PG/14/59/31000, RG/14/1/30588, P47352/Centre for Regenerative Medicine]; BIRAX [grant number 04BX14CDLG]; the Medical Research Council [grant number MR/M017354/1, G0801098]; and the Heart Research UK [grant number TRP01/12]. R.P.D. was supported by a H2020 European Research Council (ERC) Starting Grant (STEMCARDIORISK; grant agreement #638030), and a VIDI fellowship from the Netherlands Organisation for Scientific Research (Nederlandse Organisatie voor Wetenschappelijk Onderzoek; ILLUMINATE; #91715303). C.L.M. and B.J.M. were supported by the Netherlands Science Foundation (NWO) under the Gravitation Grant 'NOCI' (# 024.003.001).

## Author contributions

B.J.M. designed and carried out the experiments, designed and developed the analysis algorithm, developed the TTM system, performed the analysis and wrote the manuscript. A.K. carried out the drug screening experiments and contributed to the manuscript. L.S. carried out the MEA experiments, aided in development of the TTM system and contributed to the manuscript. C.D. organized the drug screening experiments. R.P.D. was instrumental in setting up the TTM system and contributed to the manuscript. TE co-developed the analysis algorithm and aided in interpreting the results. L.T. designed and developed the TTM system, co-developed the analysis algorithm and aided in interpreting the results. C.L.M. supervised the project and wrote the manuscript.

## Competing interests

C.L.M. is co-founder of Pluriomics bv (now Ncardia bv). The remaining authors declare no competing interests.
