## [Peer Review File · Nature Communications]

Reviewers' comments:

Reviewer #1 (Remarks to the Author):

This study by van Meer et al describe a method to simultaneously measure action potential, calcium and contraction of hiPSC-CM. Basically, the authors built a high speed optical system that can capture three fluorescent dye-based transients using commercially available components. As the author state, simultaneous measurement of two transients have been well established using fluorescent sensors, or less calcium and voltage fluorescent reporters. Meanwhile, the authors' group has reported the contractile measurement of hiPSC-CM with MUSCLEMOTION software and the CellMask Deep Red staining. The major significance of this manuscript would be using the combined measurement of these three types of transients to reveal the mechanisms underlying contractile responses of hiPSC-CM. Thus, what is new in this manuscript is the analysis algorithm that can access the mechanisms of action (MOAs) of different drugs. Overall, this work is well designed and conducted, providing novel tools to access the cardiotoxicity and efficacy of compounds.

Specific concerns:

1. Although the system can indicated the correct MOAs of 10 of 12 drugs, it is more a descriptive model and predictor because there was no new mechanisms revealed. It is misleading for the manuscript title to state that "reveals mechanisms underlying contractile responses of hiPSC-derived cardiomyocytes". I would suggest revising the title.
2. The author explained that one reason why levosimendan and pimobendan had unexpected effect is immaturity. It would be helpful to use more mature hiPSC-CMs (such as prolonged culture) to better understand the mechanisms.
3. Affiliation #4 is missing the name of institution.

Reviewer #2 (Remarks to the Author):

Recognizing the multifaceted effects of many cardioactive drugs, the authors have put forward a set of techniques specifically designed to reveal the dominant underlying mechanism of action of a drug on contraction. Five mechanisms of action were considered: increased intracellular calcium, direct myosin activation, cAMP effects, reduced calcium transient, and cardiotoxicity. 10 parameters were used to describe the effects of each drug on the action potential kinetics (amplitude, upstroke, duration, and triangulation), calcium transient kinetics (amplitude, time-to-peak, decay) and on contraction (amplitude, contraction, relaxation). Using a set of 6 calibration drugs (Aspirin, Ouabain, forskolin, Verapamil, Omecamtiv Mecarbil, Doxorubicin) the combined effects on these 10 parameters were used to assign the underlying mechanism of action of 6 other drugs (Epinephrine, Atenolol, Levosimendan, Captopril, Sunitinib, Pimobendan). Central to the study are the use of 'Probability Scores' to determine or to describe the likelihood of an increase, decrease or no drug effect on each of the 10 descriptive parameters. For each mechanism of action, the probability of matching the 10 expected effects was determined and the mechanism with the highest total score was selected as the most likely underlying mechanism of action.

To make it possible to analyze voltage, calcium and contraction kinetics simultaneously, the authors designed a microscope-based imaging system, using a high-speed camera (1000 fps), three sequentially lit LEDs (470, 565 and 656 nm) and a multi-band emission filter, to acquire independent voltage (ANINNE 6plus), calcium (Rhod 3) and contraction (CellMask Deep) signals. Because contraction results in distortion of the voltage and calcium recordings, the contraction data were used to estimate the effects of the motion artifact, which in turn allowed the voltage and calcium recordings to be 'motion corrected'.

One of the particular strengths of this study can be found in the use and generation of the probability density functions of the vehicle control condition. The authors used an experimental design of 12 drugs, with each drug condition having 6 paired recordings (5 test concentrations plus one vehicle control). Furthermore, each condition had 5 technical replicates from which three

regions of interests were selected. Therefore, this study contains an impressive 1080 (12x6x5x3) control recordings, each thoroughly described by the 10 selected parameters. Since each parameter can be described by a statistical moment containing the mean, variance, skewness and kurtosis of the parameter's distribution, there is an immense amount of data contained in the control dataset which forms the foundation of the probability scoring technique.

As datasets and the parameters that describe them become larger and larger it naturally becomes increasingly unwieldy to communicate their basic properties, especially with the space constraints of publication. However, as currently written, there is unfortunately no indication to the reader whether the probability scoring system is an effective summary of a drug's mechanism of action. Since the probability scoring system assumes that each of the 10 parameters are normally distributed, with a known mean and standard deviation, it would be greatly beneficial to the clarity of their study to provide readers with at least these two summary statistics for the control condition and at least one of the post drug conditions, if not the equivalent t-test results as well. Without this basic information, there is simply no reference for a reader to evaluate appropriateness of the probability scoring system against more conventional approach of using statistical tests to detect a significant difference between the control and drug conditions. Regardless of the appropriateness of the probability scoring system, simply knowing the expected values and range of values of these 10 parameters in the control condition is a considerable gain to the scientific community. Knowing how these values change in response to different drugs offers even more significant insight, independent from the detection abilities of the probability scoring technique.

One of the basic advantages of scoring systems, in general, lay in the encapsulation of complexity and the simplification of results interpretation. However, this encapsulation can also inadvertently obscure clarity. For example, in the 30 μM aspirin response relative to the vehicle control, there are visual changes to the action potential upstroke, action potential duration, decay of the calcium transient and contraction amplitude. While these changes may, in subsequent analysis, not be statistically significant, there is no mechanism to gain further insight to a drugs action using the probability scoring system as presented, without full access to the primary data. Comparatively, histogram-based approaches preserve more aspects of the original data – e.g. Variability of Action Potentials Within and Among Cardiac Cell Clusters Derived from Human Embryonic Stem Cells - PMC4700458

General adoption of the probability scoring system presented would be maximized by convincing readers that the selected 5 mechanisms of action have the greatest ability to separate all common cardioactive drugs into distinct categories and to show that only the direction of induced changes are relevant to their classification, rather than the magnitude of the changes observed.

Using a specific combination of changes rather than singular changes may ultimately be the key to predicting a drugs main mechanism of action. However, it also presupposes that a drug has a dominate mode of action and that no feedback mechanisms exist between voltage, calcium and contraction dynamics. Notably absent from the list of mechanisms and from the list of tested compounds are compounds that primary affect the voltage dynamics, which would subsequently affect calcium and contraction dynamics.

More Specific Points

Line 121: In the definition of $PS=$, are you dividing the $PDF(x)$ by the $PDF(\text{mean})$? When $PDF(x) \approx PDF(\text{mean})$ then the result equals 10, but if they are even slightly different what does the result go to? I'm not sure I can visualize the entire process and the resultant implications.

Line 174: "The data are then "cleaned" to remove the motion artefacts". I would prefer for the authors to give a more complete description of what was done to clean the motion artifact. They do make a reference to a ref #29 on line 185: "This yields the raw AP signal, Ca signal and the motion artefact (MA) signal, which was used to ratiometrically correct the AP and Ca signal.²⁹". But that reference uses the green and red components of a raw AP signal to 'recover' the clean AP signal. Even if it does work well, I'm not sure how one can do this if you aren't using the AP signal ratiometrically and collecting two components from the same fluorophore. In addition, the contraction is not uniform across the well nor is the degree of movement artifact. Therefore, it is not clear how much certainty there is in subtracting a generalized movement artifact when in fact

there are pixel by pixel variations. It might be possible but it is not clear to me from reading the manuscript how this was done.

Line 286: MEA Recordings of hiPSC-CMs. How concerned should one be that the QT interval as measured by MEA is ~500 ms but the AP as measured by OM is consistently ~100 ms? On that note, how concerned should one be that the AP as measured by OM is consistently ~100 ms?

Line 107: If you look at some of the references for the mechanisms of action, I'm not sure I would expect a one or two publications to contain parameters for all three types of effects (voltage, calcium, contraction). Conveniently, the assigned MOA match perfectly with their hypothesis table! This is why I make reference to 6 calibration drugs and 6 test drugs rather than 12 test drugs.

Line 227: In looking at figure 3h, Omecamtiv and Mecarbil change the voltage, calcium and contraction dynamics while the expected result are no voltage changes, no calcium changes and increased contraction amplitude and increased time of contraction but the time of contraction is shorter? But still their system classified it as Co+ Myosin?

The word "data" is plural and should be conjugated as such throughout the manuscript.

In every instance in which the term "intracellular Ca²⁺" is used, it should read cytosolic Ca²⁺.

Both Levosimendan and Pimobendan are well known Ca²⁺ sensitizing drugs in addition to them being used as PDE3-inhibiting drugs. As such I would expect the cell contraction to be greater for a given amplitude of the Ca²⁺ transient.

Reviewer #3 (Remarks to the Author):

In this article "Simultaneous measurement of action potential, calcium and contraction reveals mechanisms underlying contractile responses of hiPSC-derived cardiomyocytes", van Meer et al. present a new Triple Transient Measurement system to measure simultaneously Voltage, intracellular Calcium and contraction of hiPSC-CM cardiac cell cultures. This new system was designed under the hypothesis that individual signals may not be accurate enough to indicate the degree of arrhythmogenesis given by a drug, thus more insights in the mechanisms underlying responses to drugs can be obtained when including the information from the three signals for a combined analysis. This new system can record multiplexing the three signals at a speed of 333fps per signal (that is camera runs at 1000fps) with a large field of view of 720x720. The authors then use their new system to test and characterize 12 different drugs. Signals were induced by pacing the tissue with field stimulation at 1.2Hz.

This is a very interesting study and the system could be of great use for the identification and selection of drug's toxicity and arrhythmogenicity. This manuscript could be good fit for nature communications, however there are a few points the authors need to address, in order to show that the method is robust.

Main Comments:

As the authors mentioned, the more information we get, the better to understand the mechanism and for that we need to make sure that the signals they obtain are the real ones. The main problem I have with the methodology is that the authors have not convinced me that there is no cross talk between the signals. Something that is crucial when trying to really identify differences in the three signals by drug effects.

In Figure S1. The authors claim that the spectra of Rhod-3 and ANNINE-6 are not available, but that is not the case.

Spectra of Rhod-3 is available on Thermo-Fisher online

<https://www.thermofisher.com/order/catalog/product/R10145>

Spectra of ANNINE-6plus is published in detail as 2D excitation-emission map (one of the most extensive scanned dyes) in the original article Fromherz P, Hübener G, Kuhn B, Hinner MJ.

ANNINE-6plus, a voltage-sensitive dye with good solubility, strong membrane binding and high sensitivity. *Eur Biophys J.* 2007;37(4):509-14.

Also, Di-4-ANNEPS spectra cannot be used as a substitute for ANNINE-6plus spectra to illustrate excitation and emission spectra, what is the validation behind constructing a method with no cross talk using information from completely different dyes? Exc/Emission spectra of ANNINE-6plus has excitation/emission spectra shifted towards shorter wavelength for about 100nm, which is way too much difference.

In this study it is necessary to show that there is no cross talk between the dyes as a core foundation of the method as otherwise results are quite speculative, changes in V_m will show up changes in Ca signal or vice versa. It is also important that emphasize that separate excitation or emission spectra of the dyes is NOT enough to validate that there is no cross-talk between the dyes as spectra available in catalogs is often obtained with dyes bounded to artificial membranes and vesicles and could be quite different to that of dyes bounded to real tissue.

Excitation wavelengths for three different dyes. ANNINE-6plus 470nm, Rhod-3 565nm, CellMask Deep Red 656nm. Based on available spectra, not disclosed by the authors (while data is available).

470nm will also excite Rhod-3

565nm will also excite CellMask

There is no information about what LEDs are used to excite? These cannot be single wavelength excitations, they have a distribution, therefore information about excitation filters, band-pass, OD numbers etc have to be disclosed as well in the system's method section.

In summary. The authors fail to show that there is no cross talk between the dyes, there is no experimental validating procedure. Figure, S1 shows significant cross talk between the dyes which may be even more when using the correct spectrum. The authors need to quantify how much cross talk exist between their signals and show that it is minimal.

Other comments:

One of the main points of the paper is to compare the combined signals to the separate ones. There is summary in Table 2 comparing these two different methods, but there is lack of any quantifying measure.

Furthermore, the use of hiPSC-derived cardiomyocytes is currently used by several groups to study drug effects, the authors need to comment on other studies and how their system does actually improve results compared to others. For example with Multimodal on-axis platform for all-optical electrophysiology with near-infrared probes in human stem-cell-derived cardiomyocytes by Aleksandra Klimas, Gloria Ortiz, Steven Boggess, Evan W. Miller, Emilia Entcheva. And The Use of Ratiometric Fluorescence Measurements of the Voltage Sensitive Dye Di-4-ANEPPS to Examine Action Potential Characteristics and Drug Effects on Human Induced Pluripotent Stem Cell-Derived Cardiomyocytes M. P. Hortigon-Vinagre, V. Zamora, F. L. Burton, J. Green, G. A. Gintant, and G. L. Smith

The equation used for PDF is a Gaussian but the - sign on the equation blends with the fraction line, please make sure it is visible. In the results line 205, how exactly the algorithm was applied to the data? How MOA is ultimately assigned? How is this all related to probability scores defined by the PDF function?

The authors emphasize the new system to record V-Ca-Co simultaneously and possibility to be used in complex multicellular and 3D structures, recently a system to also measure the three signals in 3D has also been published in Nature and thus the authors should mention how would it compare with that other method. See Electromechanical vortex filaments during cardiac fibrillation by Christoph et al.

What is the strength of the field stimulation? And how it compares to threshold stimulation? How does it affects the cells after repetitive stimulations? A new optical stimulation system has been also described recently and may be mentioned and compared to this system (see Kilmas paper

above) and Light-Activated Dynamic Clamp Using iPSC-Derived Cardiomyocytes by Quach, Krogh-Madsen, Entcheva and Christini. Where they use also different protocol to account for adaptation, compared to here using only a single cycle length of 1.2Hz. This limits the range for the drug studies as often arrhythmic effects appear at a range of cycle lengths.

Table 1. shows triangulation effects, but there is no discussion about this in the text. Triangulation is a recognized hallmark of proarrhythmia. And should be cited for example Zamora paper above and Suppression of alternans and conduction blocks despite steep APD restitution: electrotonic, memory, and conduction velocity restitution effects, by Cherry et al.

Supplemental Figure 2. Authors aimed to show that the dyes do not induce any changes in baseline electrophysiological response. Supplemental Figure 2 is a weak proof. Changes in QT or RR interval are presumably obtained with the slow pacing, 1.2Hz. Even so, slow pacing showing no changes in QT or RR are not a good predictor. QT and RR measurements may be done at substantial higher pacing cycle length where alternans could appear, or at least comment on the limitation.

Moreover, a statistical analysis should be use here to show that there is no change in baseline by comparing means and variances, for example with ANNOVA test. As is, the figure shows only inferential statistics and therefore is subjected to interpretation.

The authors explain that ratiometry is used to correct for the AP and Ca signals from motion (ref 29). There is a significant number of publications of ratio-metric motion correction. They are unfortunately misinterpreted and error still propagates in the published literature. Ratiometric measurements "cleans" signal so that there is not apparent visible motion, however the method is not able to subtract motion if motion is across many pixels. Please address this problem and estimate the number of pixels used when measuring the Voltage and Calcium signal.

Also the authors mention using multiple events to average the signal, this is called ensemble averaging or staking and has been used by many when the signals have reached steady state (probably cite "Robust framework for quantitative analysis of optical mapping signals without filtering" by Uzelac and Fenton) and mention how many events were used.

Reviewers' comments:

Reviewer #1 (Remarks to the Author):

This study by van Meer et al describe a method to simultaneously measure action potential, calcium and contraction of hiPSC-CM. Basically, the authors built a high speed optical system that can capture three fluorescent dye-based transients using commercially available components. As the author state, simultaneous measurement of two transients have been well established using fluorescent sensors, or less calcium and voltage fluorescent reporters. Meanwhile, the authors' group has reported the contractile measurement of hiPSC-CM with MUSCLEMOTION software and the CellMask Deep Red staining. The major significance of this manuscript would be using the combined measurement of these three types of transients to reveal the mechanisms underlying contractile responses of hiPSC-CM. Thus, what is new in this manuscript is the analysis algorithm that can access the mechanisms of action (MOAs) of different drugs. Overall, this work is well designed and conducted, providing novel tools to access the cardiotoxicity and efficacy of compounds.

We thank the reviewer for appreciating our manuscript and for understanding the potential of our tool for analysis of cardiotoxicity.

Specific concerns:

1. Although the system can indicated the correct MOAs of 10 of 12 drugs, it is more a descriptive model and predictor because there was no new mechanisms revealed. It is misleading for the manuscript title to state that “reveals mechanisms underlying contractile responses of hiPSC-derived cardiomyocytes”. I would suggest revising the title.

We understand possible confusion our title could create and this is not our intention. Therefore, we have changed it to

"Simultaneous measurement of excitation-contraction coupling parameters identifies mechanisms underlying contractile responses of hiPSC-derived cardiomyocytes"

2. The author explained that one reason why levosimendan and pimobendan had unexpected effect is immaturity. It would be helpful to use more mature hiPSC-CMs (such as prolonged culture) to better understand the mechanisms.

We appreciate the suggestion of the reviewer and we always endeavor to produce the most mature hiPSC-CMs for our experiments. In fact, this is one of the core research areas of all authors and we have multiple publications in this area reporting our advances (e.g. PMID: 28279973, PMID: 26489474, PMID: 26209647, PMID: 28450366). Despite all efforts, however, 2D cultures of hPSC-CMs currently fall short of maturity compared to 3D cultures and engineered tissues, even after prolonged (months of) culture. Simultaneous measurement in 3D cultures will be a next step but is not yet possible in our current setup. To make the study as reproducible as possible without batch-to-batch differences in the cells used we choose commercially available Pluricytes and followed the instructions of the manufacturer for timing of the assay. This would allow other users to use the system without further development.

Pluricytes are cultured in a CM medium which, according to the manufacturer, should contain all the required factors for the maintenance of mature CMs. Thus, prolonged culture conditions would not solve the issue with levosimendan and pimobendane; conversely, this is an additional indication that maturation strategies should continue to be pursued particularly by commercial players.

3. Affiliation #4 is missing the name of institution.

We thank the reviewer for noticing this and have added the name of the institution.

Reviewer #2 (Remarks to the Author):

Recognizing the multifaceted effects of many cardioactive drugs, the authors have put forward a set of techniques specifically designed to reveal the dominant underlying mechanism of action of a drug on contraction. Five mechanisms of action were considered: increased intracellular calcium, direct myosin activation, cAMP effects, reduced calcium transient, and cardiotoxicity. 10 parameters were used to describe the effects of each drug on the action potential kinetics (amplitude, upstroke, duration, and triangulation), calcium transient kinetics (amplitude, time-to-peak, decay) and on contraction (amplitude, contraction, relaxation).

Using a set of 6 calibration drugs (Aspirin, Ouabain, forskolin, Verapamil, Omecantiv Mecarbil, Doxorubicin) the combined effects on these 10 parameters were used to assign the underlying mechanism of action of 6 other drugs (Epinephrine, Atenolol, Levosimendan, Captopril, Sunitinib, Pimobendan). Central to the study are the use of 'Probability Scores' to determine or to describe the likelihood of an increase, decrease or no drug effect on each of the 10 descriptive parameters. For each mechanism of action, the probability of matching the 10 expected effects was determined and the mechanism with the highest total score was selected as the most likely underlying mechanism of action.

To make it possible to analyze voltage, calcium and contraction kinetics simultaneously, the authors designed a microscope-based imaging system, using a high-speed camera (1000 fps), three sequentially lit LEDs (470, 565 and 656 nm) and a multi-band emission filter, to acquire independent voltage (ANINNE 6plus), calcium (Rhod 3) and contraction (CellMask Deep) signals. Because contraction results in distortion of the voltage and calcium recordings, the contraction data were used to estimate the effects of the motion artifact, which in turn allowed the voltage and calcium recordings to be 'motion corrected'.

One of the particular strengths of this study can be found in the use and generation of the probability density functions of the vehicle control condition. The authors used an experimental design of 12 drugs, with each drug condition having 6 paired recordings (5 test concentrations plus one vehicle control). Furthermore, each condition had 5 technical replicates from which three regions of interests were selected. Therefore, this study contains an impressive 1080 (12x6x5x3) control recordings, each thoroughly described by the 10

selected parameters. Since each parameter can be described by a statistical moment containing the mean, variance, skewness and kurtosis of the parameter's distribution, there is an immense amount of data contained in the control dataset which forms the foundation of the probability scoring technique.

As datasets and the parameters that describe them become larger and larger it naturally becomes increasingly unwieldy to communicate their basic properties, especially with the space constraints of publication. However, as currently written, there is unfortunately no indication to the reader whether the probability scoring system is an effective summary of a drug's mechanism of action. Since the probability scoring system assumes that each of the 10 parameters are normally distributed, with a known mean and standard deviation, it would be greatly beneficial to the clarity of their study to provide readers with at least these two summary statistics for the control condition and at least one of the post drug conditions, if not the equivalent t-test results as well. Without this basic information, there is simply no reference for a reader to evaluate appropriateness of the probability scoring system against more conventional approach of using statistical tests to detect a significant difference between the control and drug conditions. Regardless of the appropriateness of the probability scoring system, simply knowing the expected values and range of values of these 10 parameters in the control condition is a considerable gain to the scientific community. Knowing how these values change in response to different drugs offers even more significant insight, independent from the detection abilities of the probability scoring technique.

We thank the reviewer for the constructive criticism and comments. Indeed, we fully agree the comparison to 'standard' statistical parameters would help to understand the added benefit of the software tool. In our revised manuscript we have endeavored to present this data in a meaningful way to the reader (supplementary table 2) and we believe that – despite the large amount of data –we have succeeded in incorporating a useful summary. Although there may be other options to make it easier to compare the different methods, we believe this way strengthens the appropriateness of our scoring system. Moreover, for the interested reader we have developed an online tool to browse through the data and compare our method to conventional statistics. This is available at: <https://bjvanmeer.shinyapps.io/TTM-algorithm/>. In the manuscript we have included:

“Since the dataset generated is too large to include in standard figures, an interactive online tool was developed to give readers access to all data and both qualitative and quantitative insight in the data and analysis. In addition, the tool allows comparison to traditional statistics. The tool is available at <https://bjvanmeer.shinyapps.io/TTM-algorithm/>.”

In addition, we have now included the following text in our manuscript:

“A comparison with conventional statistical methods per parameter per drug per dose can be found in Supplementary Table 2. It is evident that the algorithm is typically not triggered by small changes that might not be physiologically relevant (e.g. captopril at 1 μ M) while it does indicate a relevant change in certain cases although no statistical differences are evident in the individual parameters (e.g. omeamtiv mecarbil at 0.03 and 0.3 μ M). In general, small

‘significant’ changes in single parameters might not be physiologically meaningful while a combined system of changes does indicate a relevant cell response.”

One of the basic advantages of scoring systems, in general, lay in the encapsulation of complexity and the simplification of results interpretation. However, this encapsulation can also inadvertently obscure clarity. For example, in the 30 μ M aspirin response relative to the vehicle control, there are visual changes to the action potential upstroke, action potential duration, decay of the calcium transient and contraction amplitude. While these changes may, in subsequent analysis, not be statistically significant, there is no mechanism to gain further insight to a drugs action using the probability scoring system as presented, without full access to the primary data. Comparatively, histogram-based approaches preserve more aspects of the original data – e.g. Variability of Action Potentials Within and Among Cardiac Cell Clusters Derived from Human Embryonic Stem Cells - PMC4700458

We appreciate the point that full insight in the data is hard to get from the current scoring system. While this is – as the reviewer mentions – inadvertently a consequence of the interpretation by the scoring system we would like to enable the reader interested in the underlying details to be able to evaluate these. We thank the reviewer for pointing out the histogram-based approach used by Zhu et al. which is very useful in their analysis of variability of large populations of hESC-CMs. However, we believe in our case showing the complete concentration-response curves for all tested drugs gives even more detailed insight to the reader or end-user. Therefor we have added all data in this format to the online tool mentioned above that accompanies this paper.

General adoption of the probability scoring system presented would be maximized by convincing readers that the selected 5 mechanisms of action have the greatest ability to separate all common cardioactive drugs into distinct categories and to show that only the direction of induced changes are relevant to their classification, rather than the magnitude of the changes observed.

We thank the reviewer for this advice and have added a new figure describing the qualitative interaction of the most important aspects of cardiomyocyte contractility in response to drugs (new Figure 1). We use this to explain to the reader the rationale of the different mechanisms and why they are most important in identification of contraction-modulating drugs. In addition, we provide the reader with many examples of drugs that fall into these categories. This way, we believe it is now clear to the reader that these mechanisms of action are the most important in drug classification for cardiomyocyte contractility.

Using a specific combination of changes rather than singular changes may ultimately be the key to predicting a drugs main mechanism of action. However, it also presupposes that a drug has a dominate mode of action and that no feedback mechanisms exist between voltage, calcium and contraction dynamics. Notably absent from the list of mechanisms and from the

list of tested compounds are compounds that primary affect the voltage dynamics, which would subsequently affect calcium and contraction dynamics.

Again, the reviewer has a good point and we appreciate the comment. We have deliberately chosen to look only at contraction-modulating drugs since this type of drug is of high interest in the field and other scoring systems exist for drugs affecting voltage dynamics that can be studied using simpler tools (e.g. MEAs or VSDs only). However, our intention is to incorporate these types of drugs into our algorithm in the future and thus make the scoring system even more widely applicable.

More Specific Points

Line 121: In the definition of $PS=$, are you dividing the $PDF(x)$ by the $PDF(\text{mean})$? When $PDF(x) \approx PDF(\text{mean})$ then the result equals 10, but if they are even slightly different what does the result go to? I'm not sure I can visualize the entire process and the resultant implications.

This depends on the shape of the PDF, which in turn depends on the vehicle values. Since the PDF function of $PDF(x)$ and $PDF(\text{mean})$ are the same function, we merely express $PDF(x)$ normalized to $PDF(\text{mean})$. The area under the curve of the PDF is normally set to 1. However, since we are not working with chance (between 0-1), but with a scoring factor, we want to have in a range between 0-10 thus we need to set the top of the PDF (which is $PDF(\text{mean})$) at 10. We do this by dividing $PDF(x)$ by $PDF(\text{mean})$ (i.e. normalizing to 1) and then multiplying by 10.

To make it clearer to the reader we have modified the following sentences:
“(…), the PDF was normalized to the mean and multiplied by 10 to have a range of 0-10. The resulting function was divided in three (…).”

Line 174: “The data are then “cleaned” to remove the motion artefacts”. I would prefer for the authors to give a more complete description of what was done to clean the motion artifact. They do make a reference to a ref #29 on line 185: “This yields the raw AP signal, Ca signal and the motion artefact (MA) signal, which was used to ratiometrically correct the AP and Ca signal.²⁹”. But that reference uses the green and red components of a raw AP signal to ‘recover’ the clean AP signal. Even if it does work well, I’m not sure how one can do this if you aren’t using the AP signal ratiometrically and collecting two components from the same fluorophore. In addition, the contraction is not uniform across the well nor is the degree of movement artifact. Therefore, it is not clear how much certainty there is in subtracting a generalized movement artifact when in fact there are pixel by pixel variations. It might be possible but it is not clear to me from reading the manuscript how this was done.

We thank the reviewer for this comment and we appreciate this might not have been clear from the explanation given in the manuscript. Indeed, we refer to use of a different dye that has a green and red component and can thus be ratiometrically imaged. However, although we do not use the same dye, we obtain the same data. Apart from our non-ratiometric voltage dye signal we also have the signal from our CellMask Deep Red dye which is co-localized with the voltage dye in the membrane of the cell. This signal does not change in fluorescence

intensity, but only in the image localization due to contraction of the cardiomyocytes. Since this signal is recorded at the same time and location as the VSD signal using our high speed TTM system, we can correlate these signals and use them in the same way as described in the referenced paper.

To make this procedure clearer to the reader we have modified the sentence as follows: “(...) and the motion artefact (MA) signal. Since CellMask Deep Red (i.e. the fluorophore that is the origin of the MA signal) is co-localized with the VSD in the membrane of the hiPSC-CMs and recorded simultaneously with the VSD signal, we could use the MA signal for ratiometric correction of the AP and Ca signal.”

Line 286: MEA Recordings of hiPSC-CMs. How concerned should one be that the QT interval as measured by MEA is ~500 ms but the AP as measured by OM is consistently ~100 ms? On that note, how concerned should one be that the AP as measured by OM is consistently ~100 ms?

We thank the reviewer for pointing out this discrepancy. MEA recordings and TTM analyses were performed on different iPSC lines. Specifically, MEA measurements were performed on a Jervell and Lange-Nielsen Syndrome (JLNS) cell line from Zhang et al., 2014 which, as expected, exhibited QT and RR values similar to those previously published (Sala et al., EMBO Mol Med, 2016). This cell line is derived from patients with a severe arrhythmogenic disorder and, at least theoretically, it would have been more sensitive to any cytotoxic effect of the fluorescent dyes. This did not occur and thus allowed us to have confidence in the reliability of our protocol both for healthy and disease lines.

On the other hand, all the measurements for the TTM were performed on a commercially available healthy control line. This explains the discrepancy between QT and APD values. We apologize for the error which we have acknowledged in the manuscript, as follows:

“To investigate the extent to which the optical sensors affected baseline electrophysiological responses, hiPSC-CMs previously derived from patients with Jervell and Lange-Nielsen Syndrome (JLNS) were plated on MEAs and measured spontaneously before and after 18-20 minutes incubation (Supplemental Figure 2). Inferential statistics with ANOVA was performed and no significant effect of the sensors was found. In addition, values found were similar to those previously reported.”

Line 107: If you look at some of the references for the mechanisms of action, I'm not sure I would expect a one or two publications to contain parameters for all three types of effects (voltage, calcium, contraction). Conveniently, the assigned MOA match perfectly with their hypothesis table! This is why I make reference to 6 calibration drugs and 6 test drugs rather than 12 test drugs.

The reviewer is correct that this data is not fully described in the references we initially listed. We have therefore added more references to provide a clearer overview of the literature on the different mechanism. Our original intention for including a limited number of references was to simplify the concept for the reader, but we fully agree we should provide a more extensive list of references to justify our hypothesis table. Please find a more extensively documented table 1 in our revised manuscript.

Line 227: In looking at figure 3h, Omecamtiv and Mecarbil change the voltage, calcium and contraction dynamics while the expected result are no voltage changes, no calcium changes and increased contraction amplitude and increased time of contraction but the time of contraction is shorter? But still their system classified it as Co^{+} Myosin?

We thank the reviewer for this comment. Interestingly, we see in the concentration-response curve (pasted below) that the contraction time is not really changed at this concentration. The amplitude is higher after addition of OM and the relaxation time has increased as well, as expected. However, as the reviewer mentions, the algorithm is able to pick it up these, more subtle, changes.

The word “data” is plural and should be conjugated as such throughout the manuscript.

We thank the reviewer and have corrected this throughout the manuscript.

In every instance in which the term “intracellular Ca^{2+} ” is used, it should read cytosolic Ca^{2+} .

We thank the reviewer and have corrected this throughout the manuscript.

Both Levosimendan and Pimobendan are well known Ca^{2+} sensitizing drugs in addition to them being used as PDE3-inhibiting drugs. As such I would expect the cell contraction to be greater for a given amplitude of the Ca^{2+} transient.

Indeed, we would have expected this effect as well but we believe hiPSC-CM immaturity is causing the lack of effect. Since we only mentioned the PDE3 effect we have now changed the text to include Ca^{2+} sensitization as well:

“Both PDE3-inhibiting drugs levosimendan and pimobendan had no effect assigned by the algorithm, whilst the opposite would be expected based on the literature, where the calcium sensitizing- and PDE3-inhibiting effects of the drugs should cause a positive inotropic effect.”

Reviewer #3 (Remarks to the Author):

In this article “Simultaneous measurement of action potential, calcium and contraction reveals mechanisms underlying contractile responses of hiPSC-derived cardiomyocytes”, van Meer et al. present a new Triple Transient Measurement system to measure simultaneously Voltage, intracellular Calcium and contraction of hiPSC-CM cardiac cell cultures. This new system was designed under the hypothesis that individual signals may not be accurate enough to indicate the degree of arrhythmogenesis given by a drug, thus more insights in the mechanisms underlying responses to drugs can be obtained when including the information from the three signals for a combined analysis. This new system can record multiplexing the three signals at a speed of 333fps per signal (that is camera runs at 1000fps) with a large field of view of 720x720. The authors then use their new system to test and characterize 12 different drugs. Signals were induced by pacing the tissue with field stimulation at 1.2Hz.

This is a very interesting study and the system could be of great use for the identification and selection of drug’s toxicity and arrhythmogenicity. This manuscript could be good fit for nature communications, however there are a few points the authors need to address, in order to show that the method is robust.

We thank the reviewer for acknowledging the potential of the study and we appreciate the reviewer’s feedback to further demonstrate the robustness of the method.

Main Comments:

As the authors mentioned, the more information we get, the better to understand the mechanism and for that we need to make sure that the signals they obtain are the real ones. The main problem I have with the methodology is that the authors have not convinced me that there is no cross talk between the signals. Something that is crucial when trying to really identify differences in the three signals by drug effects.

We understand the concern of the reviewer and have now included a supplemental figure (Supp. Fig. 3) confirming the absence of cross talk between the signals. To demonstrate that there is no crosstalk, we loaded three separate wells with each of the dyes individually and then recorded the signal in all channels upon sequential exposure to each of the LEDs for 7 seconds (cells were paced at 1.2 Hz). There is no significant illumination of the dyes when they were excited by the LEDs that were not within their excitation spectrum, confirming the absence of cross talk.

We have added the following sentence in the text:

“To test whether there was any crosstalk between the fluorescent signals of the different dyes, different wells of Pluricyte hiPSC-CMs were exclusively labelled with each of the three dyes and exposed to all three LED wavelengths individually (Supplementary Figure 3). Assessment of the fluorescence intensity revealed that ANNINE-6plus and Rhod 3 signals were only present in response to 470 and 560 nm wavelengths, respectively. CellMask Deep Red showed a modulating image intensity due to contraction usable for MUSCELMOTION analysis only at 656 nm.”

In Figure S1. The authors claim that the spectra of Rhod-3 and ANNINE-6 are not available, but that is not the case.

Spectra of Rhod-3 is available on Thermo-Fisher online

<https://www.thermofisher.com/order/catalog/product/R10145>

Spectra of ANNINE-6plus is published in detail as 2D excitation-emission map (one of the most extensive scanned dyes) in the original article Fromherz P, Hübener G, Kuhn B, Hinner MJ. ANNINE-6plus, a voltage-sensitive dye with good solubility, strong membrane binding and high sensitivity. Eur Biophys J. 2007;37(4):509-14.

Also, Di-4-ANNEPS spectra cannot be used as a substitute for ANNINE-6plus spectra to illustrate excitation and emission spectra, what is the validation behind constructing a method with no cross talk using information from completely different dyes? Exc/Emission spectra of ANNINE-6plus has excitation/emission spectra shifted towards shorter wavelength for about 100nm, which is way too much difference.

We apologize for our inaccuracy in the information of the spectra. The reviewer is completely right and while we were at first limited by the spectra that we could find in digital format we have now digitalized the available spectra of ANNINE-6plus and Rhod-3. Both spectra are incorporated in the Supplementary Figure S1. We thank the reviewer for this comment, because the reader can now see more clearly that the excitation spectra of the different dyes have almost no overlapping area.

In this study it is necessary to show that there is no-cross talk between the dyes as a core foundation of the method as otherwise results are quite speculative, changes in V_m will show up changes in Ca signal or vice versa. It is also important that emphasize that separate excitation or emission spectra of the dyes is NOT enough to validate that there is no cross-talk between the dyes as spectra available in catalogs is often obtained with dyes bounded to artificial membranes and vesicles and could be quite different to that of dyes bounded to real tissue.

We completely agree with the reviewer and have performed an extra experiment as described in our response above.

Excitation wavelengths for three different dyes. ANNINE-6plus 470nm, Rhod-3 565nm, CellMask Deep Red 656nm. Based on available spectra, not disclosed by the authors (while data is available).

470nm will also excite Rhod-3

565nm will also excite CellMask

There is no information about what LEDs are used to excite? These cannot be single wavelength excitations, they have a distribution, therefore information about excitation filters, band-pass, OD numbers etc have to be disclosed as well in the system's method section.

We thank the reviewer for pointing out this omission and have included the information requested in full in the methods section. It now reads:

“An Eclipse Ti microscope (Nikon) was fitted with a 3 high-power LED system (Mightex; 470 nm, 560 nm and 656 nm) including collimators and excitation filters (Semrock; 470±14nm, 544±12nm, 650±7.5nm).”

In summary. The authors fail to show that there is no cross talk between the dyes, there is no experimental validating procedure. Figure, S1 shows significant cross talk between the dyes which may be even more when using the correct spectrum. The authors need to quantify how much cross talk exist between their signals and show that it is minimal.

We agree with the reviewer and performed the extra experiment as described in our response above to address this. As we clearly show, there is no crosstalk between the different channels. As suggested by the reviewer, we have also added more information on the spectra of the dyes and LEDs used in our assay.

Other comments:

One of the main points of the paper is to compare the combined signals to the separate ones. There is summary in Table 2 comparing these two different methods, but there is lack of any quantifying measure.

The algorithm reports a most likely scenario and it is not possible to give an absolute metric since it is determined by the individual vehicle populations. Because of this, we chose not to display numbers since it is not relevant to compare different situations, experiments and configurations with each other. We believe that including the numbers would make readers more confused. However, to give the reader access to the quantitative data if they wish to examine it, we have developed an online tool where all data can be accessed both qualitatively and quantitatively in comparison to standard statistics. This is available at: <https://bjvanmeer.shinyapps.io/TTM-algorithm/>

Furthermore, the use of hiPSC-derived cardiomyocytes is currently used by several groups to study drug effects, the authors need to comment on other studies and how their system does actually improve results compared to others. For example, with Multimodal on-axis platform for all-optical electrophysiology with near-infrared probes in human stem-cell-derived cardiomyocytes by Aleksandra Klimas, Gloria Ortiz, Steven Boggess, Evan W. Miller, Emilia Entcheva. And The Use of Ratiometric Fluorescence Measurements of the Voltage Sensitive Dye Di-4-ANEPPS to Examine Action Potential Characteristics and Drug Effects on Human Induced Pluripotent Stem Cell-Derived Cardiomyocytes M. P. Hortigon-Vinagre, V. Zamora, F. L. Burton, J. Green, G. A. Gintant, and G. L. Smith

We thank the reviewer for raising this point and have added a more detailed comparison to other platforms in our manuscript:

“Other platforms that use multi parameter assessments in hPSC-CMs exist but differ from the TTM system. The system developed by Klimas et al. relies on optogenetics and therefore requires cell lines in which the optogenetic actuators have been transfected or stably inserted, adding significant complexity and decreasing flexibility. Furthermore while the channelrhodopsin2 enables the cells to be paced, it has to be controlled by light pulses which limits the time for fluorescence recording and therefore only allows simultaneous measurements of two parameters (voltage and calcium) rather than all three. While the CelLOPTIQ system (Clyde Biosciences) can record all three parameters, this can only be performed sequentially and not simultaneously like the TTM system. Additionally, the CelLOPTIQ system has a lower frame rate which might limit the accuracy especially for the action potential. However, the resulting output is similar to that obtained with the TTM system, meaning that similar analysis using the hypothesis-based model for the classification of pharmacological effects might be possible. Similarly, Christoph et al. recently published a more complex method to perform accurate 3D voltage mapping. While only suitable for 3D tissues and more focused on the spatiotemporal patterns during fibrillation, the output data from this system might be suitable for the hypothesis-based algorithm.

Automated evaluation of the output data generated by different systems using the hypothesis-based algorithm would enable comparable interpretation of the results. Since the algorithm is based on the vehicle population of the system it would not be limited by differences between absolute values or percentages of change which often make head to head comparison of different techniques difficult or impossible. More generally, user bias is greatly reduced using automated interpretation. However, currently all systems still rely on non-automated manual response interpretation.”

The equation used for PDF is a Gaussian but the – sign on the equation blends with the fraction line, please make sure it is visible.

We thank the reviewer for spotting and we have fixed the issue in the text.

In the results line 205, how exactly the algorithm was applied to the data? How MOA is ultimately assigned? How is this all related to probability scores defined by the PDF function?

Apparently it is unclear how the algorithm described in the paper is applied to the data. We have therefore added the following to the results and made an extra table to give the reader more insight:

“For each drug and at each concentration, the hypothesis table was populated using the calculations of the probability scores and summarized (Supplementary Table 1). As described earlier, the highest summarized value identified the MOA.”

The authors emphasize the new system to record V-Ca-Co simultaneously and possibility to be used in complex multicellular and 3D structures, recently a system to also measure the three signals in 3D has also been published in Nature and thus the authors should mention

how would it compare with that other method. See Electromechanical vortex filaments during cardiac fibrillation by Christoph et al.

We thank the reviewer for the suggestion and we have included a discussion on this system in the text. Please see our response above.

What is the strength of the field stimulation? And how it compares to threshold stimulation? How does it affect the cells after repetitive stimulations? A new optical stimulation system has been also described recently and may be mentioned and compared to this system (see Kilmas paper above) and Light-Activated Dynamic Clamp Using iPSC-Derived Cardiomyocytes by Quach, Krogh-Madsen, Entcheva and Christini. Where they use also different protocol to account for adaptation, compared to here using only a single cycle length of 1.2Hz. This limits the range for the drug studies as often arrhythmic effects appear at a range of cycle lengths.

We thank the reviewer for these comments. Although optogenetics and light-activated ion channels are undoubtedly a powerful tool to overcome current limitations of hPSC models, we did not want to incorporate them in our setup at this time. Every optogenetic actuator has an excitation wavelength that allows the particular ion channel class to open. In our setup, this would limit the available wavelength spectrum for the measurements. In addition, we currently have not focused on long term measurements but more acute responses.

The choice of the stimulation frequency is a trade-off. Chronotropic effects will indeed not be detected (easily) while using a fixed pacing frequency. While using an adaptive frequency might be useful for that, it complicates the comparison of vehicle measurements with each other since the calculation to correct for cycle length are not ideal for electrophysiological properties and not available for calcium and contraction. We chose 1.2 Hz to ensure all cells would respond and that no variation in stimulation frequency was required in the vehicle measurements. We understand the reviewer's comment and appreciate that using a low frequency would help in detecting increased contraction amplitudes, and while we feel many details of this discussion are outside of the scope of our work we have included the following sentence to make our choice clearer to the reader:

“Our algorithm uses vehicle measurements to define the baseline population, so a fixed pacing frequency of 1.2 Hz was chosen for all measurements instead of an adaptive frequency. The latter would have required frequency correction for all parameters but the mathematical models for such correction are imperfect for action potentials and unavailable for calcium and contraction.”

Regarding the stimulation method, we have also included in the revised manuscript the following:

“hiPSC-CMs were stimulated using a method developed in-house to produce a block pulse that was chosen to be as short as possible (2-10 ms) and used a voltage as low as possible (10-18 V) while still activating the cells. The settings were kept consistent while measuring under baseline conditions and after drug addition. All wells were measured, and thus stimulated, only twice and no changes were seen in susceptibility of response to the field stimulation during the second measurement. A 50% medium change during drug addition minimized electrolysis effects.”

Table 1. shows triangulation effects, but there is no discussion about this in the text. Triangulation is a recognized hallmark of proarrhythmia. And should be cited for example Zamora paper above and Suppression of alternans and conduction blocks despite steep APD restitution: electrotonic, memory, and conduction velocity restitution effects, by Cherry et al.

We thank the reviewer for this observation. We have expanded the section concerning AP triangulation and cited the reference provided.

The manuscript now reads:

“While most kinetic parameters in Table 1 are simple measurements of parts of the waveform, action potential triangulation is calculated as the difference between APD at 30% of repolarization and APD at 90% of repolarization. Triangulation is instrumental in detecting proarrhythmic effects.”

Supplemental Figure 2. Authors aimed to show that the dyes do not induce any changes in baseline electrophysiological response. Supplemental Figure 2 is a weak proof. Changes in QT or RR interval are presumably obtained with the slow pacing, 1.2Hz. Even so, slow pacing showing no changes in QT or RR are not a good predictor. QT and RR measurements may be done at substantial higher pacing cycle length where alternans could appear, or at least comment on the limitation.

Moreover, a statistical analysis should be use here to show that there is no change in baseline by comparing means and variances, for example with ANNOVA test. As is, the figure shows only inferential statistics and therefore is subjected to interpretation.

We thank the reviewer for giving us the opportunity to discuss this point further. In Supplemental Figure 2, we aimed to demonstrate that the dye combination was not causing either cardiotoxic effects or abnormal changes in the baseline electrophysiological properties of hiPSC-CMs. Our concerns were i) the potential cardiotoxic effect of the dye combination, which would result in altered FP parameters; ii) the potentially increased membrane fluidity, which could affect ion channel properties and, thus, both depolarization (sodium channel complex formation) and repolarization (hERG tetramer formation, IKs complex formation); (For hERG: PMID: 20510171; For sodium channels: PMID: 15111647) and; iii) the potentially high Ca²⁺-buffering effect by the Ca²⁺-binding dye, which would have significantly shortened the QT interval and altered the RR interval.

To evaluate this, we used one of the most sensitive cell lines we had in our biobank, which carries the (homozygous) KCNQ1R594Q mutation, causing the Jervell and Lange-Nielsen Syndrome, one of the most severe forms of Long QT Syndrome (Zhang et al., 2014). We hypothesized that this cell line would reveal any potential effect induced by the dye combination on the baseline electrophysiological properties.

Interestingly, no changes were observed in any of the parameters analyzed, indicating not only that the dye combination (within the proposed experimental settings), does not affect cell viability, but also that the electrophysiological parameters are unaffected. As we were

interested in the potential effects of the dye combination on the beat rate (RR interval) of the cells, we used spontaneously beating CMs (not paced at 1.2 Hz). Our data confirmed that the dyes had no effects on the hiPSC-CMs' beating rhythm.

As requested, we have performed inferential statistics using ANOVA and did not find any difference either in QT- or RR Intervals after incubation with the dyes. Thus, we can confirm that the incubation protocol used here does not alter the baseline electrophysiological properties of the hiPSC-CMs and that we consider this dye combination safe in the conditions reported in this paper.

For clarity to the reader we have modified the text as follows:

“To investigate the extent to which the optical sensors affected baseline electrophysiological responses, hiPSC-CMs previously derived from patients with Jervell and Lange-Nielsen Syndrome (JLNS)³⁹ were plated on MEAs and the spontaneous electrophysiological response was measured before and after incubation with the dyes (Supplemental Figure 2). Inferential statistics with ANOVA was performed and no significant effect of the sensors was found. In addition, values found were similar to those previously reported.”

The authors explain that ratiometry is used to correct for the AP and Ca signals from motion (ref 29). There is a significant number of publications of ratio-metric motion correction. They are unfortunately misinterpreted and error still propagates in the published literature. Ratiometric measurements “cleans” signal so that there is not apparent visible motion, however the method is not able to subtract motion if motion is across many pixels. Please address this problem and estimate the number of pixels used when measuring the Voltage and Calcium signal.

We thank the reviewer for raising this point. Indeed, ratiometric motion correction used to clean the traces does not work correctly in the presence of large movements, i.e. with large portions of the cell/monolayer moving in and out the field of view. As can be seen from the setup figure, our analysis evaluates movement artifact after retrieval of the fluorescent signal which is captured using all 720x720 (=518400 pixels). This ratiometric approach based on literature reduces the small displacements present in the field of view. All hiPSC-CMs that were measured were attached to the bottom of the well plate which significantly limits their (large) movements. In case of 3D tissues, compensation will become more difficult due to the large free movements of the tissue. Initially, we wanted to develop a control using blebbistatin but we found the effects on the electrophysiological properties were too large to be useful (also described in literature: PMID: 23291912). We strongly believe this is currently the best method to correct for the motion artifact and that other options are less optimal.

Also the authors mention using multiple events to average the signal, this is called ensemble averaging or staking and has been used by many when the signals have reached steady state (probably cite “Robust framework for quantitative analysis of optical mapping signals without filtering” by Uzelac and Fenton) and mention how many events were used.

We thank the reviewer for this suggestion. We have integrated the proposed reference in the text. In addition, we have added the number of events (5) in the text.

Reviewers' comments:

Reviewer #2 (Remarks to the Author):

This study has been conducted by an excellent team of researchers known for their cutting edge work on hiPSC-CMs.

The problem that they are tackling is an extremely important one and deserves rigorous investigation. The manuscript has improved as a response to the comments from the reviewers. However having said that I still have concerns which are listed below.

The primary reservations on the first submissions were, the lack of clarity relating the 'raw' effect of a drug and the resultant probability score. I think they've moved towards more clarity by including the online data tool/supplement (i.e. shinyapps.io link) which contains some useful summary statistics on their basic parameters.

Most certainly, the online data tool is an important improvement and I think it deserves to be in at least the proper supplement. Essentially, however what one needs is enough clarity in the raw data to decide whether a multi-parameter scoring system can even be entertained in the context of the data quality, irrespective of any technical limitations. Because the online data tool lets you see how much spread they have in the normalized drug data, you can at least see that there is significant variability in the magnitude and direction of changes. This variability might be why traditional statistics were 'not able' to detect certain drug results and why they opted to use a multi-parameter scoring system.

In the online data tool, they included the mean/variance of their vehicle control but not of their control data. (Note that summary statistics for the control conditions was specifically requested.) However, as reported, the vehicle control as normalized to baseline/ control conditions contained (in the concentration-response graphs) a concerning amount of variability. Perhaps their vehicle control is being presented as a percentage of the baseline mean? Either way, it is hard to imagine putting much weight in a scoring system developed using cells that have vehicle controls resulting in 50% ->200% differences in contraction amplitude (i.e. doxorubicin).

If the raw values had been presented and had this level of variability, I would be inclined to give them the benefit of the doubt, but the absence of that data makes one wonder why it was omitted.

Even after normalization, which should theoretically correct for cell-to-cell variability (e.g. having different starting baseline values), there still is so much variability in the drug data. While the average response might change in one direction, the contributing measurements both get shorter and longer: the changes in mean values are small and the measurement variances are large. There is then somewhat of a philosophical question or statistical question, can you take non-statistically significant values and create an aggregate parameter to predict the mechanism of a 'maybe' drug effect? How would you then verify your hypothesis? Presumably, in a screening application, you would check from individual measurements for statistical significance, but in this case the authors already know that there isn't significance. So, then is the logic just circular?

As long as there is enough raw data being presented so that a future reader can make their own decision, really not that concerned about the authors own conclusions. However, I feel like there are lots of "for the interests of simplicity and clarity we have omitted the interesting parts of the data". Normally one uses statistically significant changes to suggest that there might be physiological significance. Here, the authors are arguing that statistically insignificant changes have physiological significance ("In general, small 'significant' changes in single parameters might not be physiologically meaningful while a combined system of changes does indicate a relevant cell response.").

Despite the 'improved clarity' with the online data tool (which is still very messy: no units etc.), the specifics of the scoring system are still as unclear as before. The logic of the scoring system may be sound, that the combination of effects may be a predictor of the mechanism of action. Perhaps, if traditional statistical tests showed the profile of 'hypothesized' responses, then maybe

there is something to work with for the use of an aggregate score. But the idea that a pattern of non-statistically significant changes can be used for the same purpose is an extra logical leap that makes the analysis of the approach very challenging.

Reviewer #4 (Remarks to the Author):

This study presents a hypothesis-based statistical algorithm and triple measurement system for highthroughput drug screening using hiPSC-CMs. The study is rigorous and results compelling. Large amount of data generated by the study is available through an online tool. The authors have been very responsive to previous reviewers, incorporating additional data to validate the approach among other improvements. There are no outstanding concerns.

Response to reviewers

Reviewer #2 (Remarks to the Author):

This study has been conducted by an excellent team of researchers known for their cutting edge work on hiPSC-CMs.

The problem that they are tackling is an extremely important one and deserves rigorous investigation. The manuscript has improved as a response to the comments from the reviewers. However having said that I still have concerns which are listed below.

We thank the reviewer for acknowledging the importance of the problem we address in this work and the improvement of the manuscript since the previous version. We agree that – with help of the reviewers – we have been able to make the manuscript even more robust. In this second revision, we believe we have strengthened the manuscript even further by addressing the new issues raised.

The primary reservations on the first submissions were, the lack of clarity relating the ‘raw’ effect of a drug and the resultant probability score. I think they’ve moved towards more clarity by including the online data tool/supplement (i.e. shinyapps.io link) which contains some useful summary statistics on their basic parameters.

We appreciate the reviewer’s comment that the modifications provided more clarity.

Most certainly, the online data tool is an important improvement and I think it deserves to be in at least the proper supplement. Essentially, however what one needs is enough clarity in the raw data to decide whether a multi-parameter scoring system can even be entertained in the context of the data quality, irrespective of any technical limitations. Because the online data tool lets you see how much spread they have in the normalized drug data, you can at least see that there is significant variability in the magnitude and direction of changes. This variability might be why traditional statistics were ‘not able’ to detect certain drug results and why they opted to use a multi-parameter scoring system.

Unfortunately, this high variability is at present still an intrinsic feature of hiPSC-CMs, and has been described in detail in multiple publications where various strategies were used in attempts to reduce it. One of these strategies includes the use of positive control wells in every treated plate to serve as quality control. One notable example of this was a publication (Kopljar et al., 2018.

<https://dx.doi.org/10.1016%2Fj.stemcr.2018.11.007>), where the authors developed a scoring system to identify cardiac hazard based on the properties of calcium transients.

Even though the use of quality control wells allowed the researchers to exclude ~ 15% of the recorded samples (they did not pass quality control), the final results do still show significant variability, with responses to the same pharmacological stimulus ranging from an increase of +50% to more than +1000% (e.g. Fig 2E). This is much greater than the range we report here.

We have added the following sentence to the manuscript to make clearer to the reader the frequently-encountered variability:

“The majority of hiPSC-CMs remain immature in most standard laboratory formats and are therefore associated with a higher variability among biological replicates. Whilst methods are emerging that can promote maturation, advanced multiplexed tools as described here can help identify immature features and suggest possible methods to address this and reduce inter-sample variability.”

In the online data tool, they included the mean/variance of their vehicle control but not of their control data. (Note that summary statistics for the control conditions was specifically requested.)

We apologize for this misunderstanding. We were under the impression the referee meant 'vehicle' data since that serves as our control for each measurement. We have integrated the baseline statistics of all measurements in the online tool. As expected, the baseline spread overall is large. We have added the following sentence to the manuscript:

Baseline statistics and raw concentration-response curves are also available to demonstrate the high inter-measurement variation and time-dependent effects, respectively, which is the reason for normalization to the vehicle.

However, as reported, the vehicle control as normalized to baseline/ control conditions contained (in the concentration-response graphs) a concerning amount of variability. Perhaps their vehicle control is being presented as a percentage of the baseline mean? Either way, it is hard to imagine putting much weight in a scoring system developed using cells that have vehicle controls resulting in 50% ->200% differences in contraction amplitude (i.e. doxorubicin).

Indeed, as we mention in the manuscript, we normalize against the vehicle data. This is because there are changes under vehicle conditions which might make data interpretation difficult for the reader. However, to make it clearer, we have now included a new set of concentration-response curves in the online tool ('concentration-response curves RAW'). In these graphs, the reader can appreciate the measurements before ('baseline') and after ('added') adding the vehicle or drug concentration. It should be kept in mind that these points are now shown as different means before and after addition of the vehicle or drug concentration but that they are paired in reality. This means that the apparently large spread is reduced because we measure exactly the same point before and after addition of the vehicle or drug concentration and calculate the relative effect (see 'concentration-response curves').

In addition, the original data table is available (as Source Data File accompanying the manuscript) for independent processing.

If the raw values had been presented and had this level of variability, I would be inclined to give them the benefit of the doubt, but the absence of that data makes one wonder why it was omitted.

As mentioned above, we have primarily put effort into developing tools to guide readers through the vast amount of data generated by our system. The raw data indeed had this level of variability which will be familiar to end users. Of note also, the raw data of the amplitudes of the three parameters are expressed in arbitrary units so cannot be directly compared to measurements in the literature, which is why we decided to present normalized data rather than raw values. However, we appreciate that the reviewer has concerns about its omission without explanation, so for transparency we will provide the source data with the manuscript. To make it easier to browse the data, we have included RAW concentration-response graphs in the online tool. We believe this will make it straightforward for readers to browse the data in many different ways and we regret that omission raised concerns with the referee. We hope that our initial intentions are now clear and believe that the re-revised manuscript and online tool now address the issue.

Even after normalization, which should theoretically correct for cell-to-cell variability (e.g. having different starting baseline values), there still is so much variability in the drug data. While the average response might change in one direction, the contributing measurements both get shorter and longer: the changes in mean values are small and the measurement variances are large. There is then somewhat of a philosophical question or statistical question, can you take non-statistically significant values and create an aggregate parameter to predict the mechanism of a

'maybe' drug effect? How would you then verify your hypothesis? Presumably, in a screening application, you would check from individual measurements for statistical significance, but in this case the authors already know that there isn't significance. So, then is the logic just circular?

We appreciate this "thought exercise" and our reasoning is as follows: we started with a theoretical model and tested it with an experimental design capable of capturing the relevant (known) responses. Verification with traditional statistics is to our knowledge impossible since these do not include the response of multiple parameters combined as we do here.

It is not correct that we know in advance there is no significance. Both options of significance or non-significance are possible. The simplest example is a concentration-response curve of contraction that goes down over 5 concentrations. At concentration 5, the data is statistically significant while at dose 3 it might not be. But the trend line does go down and, if a curve were fitted, the confidence interval might be small enough to draw conclusions. However, in many cases (e.g. if the response does not reach plateau phase) the curve cannot be fitted. We wanted to provide an alternative strategy without stating significance (which includes defined 'arbitrary' thresholds) and instead make a comparative model and verify this with experimental data. Our TTM system is intended as a (preclinical) tool to obtain reliable indications on whether and how a drug might affect one of the key parameters of the cardiac EC coupling. We aim to understand, on average, what the likely cardiomyocyte responses will be to an unknown compound before moving further down the drug screening and drug development pipelines. Therefore, the system may be used to identify and develop potential drug candidates that will be validated in the later stages in platforms with lower throughput capabilities.

As stated above, the level of variability we observe in the drug data is currently intrinsic to hiPSC-CMs and described by all researchers in the field (Knollmann, 2013, <https://www.ahajournals.org/doi/10.1161/CIRCRESAHA.112.300567>). We aimed to minimize this by using commercially differentiated hiPSC-CMs, which, in principle, should offer a more robust platform. However, even with all these precautions, we could not eliminate or control all the potential variables involved and, for these reasons, we proposed that it is better to rely more on aggregated data rather than on individual data points. More reliable QCs might be available in the future, as well as more robust and less variable cellular models. This would be welcome additions to the model here and will contribute to decreased data variability and increase the robustness of our TTM system.

As long as there is enough raw data being presented so that a future reader can make their own decision, really not that concerned about the authors own conclusions. However, I feel like there are lots of "for the interests of simplicity and clarity we have omitted the interesting parts of the data". Normally one uses statistically significant changes to suggest that there might be physiological significance. Here, the authors are arguing that statistically insignificant changes have physiological significance ("In general, small 'significant' changes in single parameters might not be physiologically meaningful while a combined system of changes does indicate a relevant cell response.").

We appreciate this comment and we have rephrased this so that readers do not have the same impression: we do not argue they never represent physiological changes but merely they might. The sentence now reads as follows:

"In some cases, small 'significant' changes in single parameters might not be physiologically meaningful while a combined system of changes does more robustly indicate a relevant cell response."

Despite the 'improved clarity' with the online data tool (which is still very messy: no units etc.), the specifics of the scoring system are still as unclear as before. The logic of the scoring system may be sound, that the combination of effects may be a predictor of the mechanism of action. Perhaps, if traditional statistical tests showed the profile of 'hypothesized' responses, then maybe there is something to work with for the use of an aggregate score. But the idea that a pattern of non-statistically significant changes can be used for the same purpose is an extra logical leap that makes the analysis of the approach very challenging.

We thank the reviewer for pointing this out and have added units more clearly in the online tool (i.e. in most cases percentages, so no actual unit). We also thank the reviewer for agreeing on the sound logic of the scoring system. As stated earlier, our intention was to demonstrate the applicability of this scoring system using well-known compounds to generate experimental data and we think the data is convincing. We believe this is the only way to demonstrate the applicability of our system since – as the referee states without suggesting an alternative – traditional statistics are not able to capture the combined response of the different parameters. We are aware of these shortcomings of traditional statistics and have proposed a way to overcome them by combining multiple simultaneously measured parameters using an algorithm based on known cardiac response.

Reviewer #4 (Remarks to the Author):

This study presents a hypothesis-based statistical algorithm and triple measurement system for highthroughput drug screening using hiPSC-CMs. The study is rigorous and results compelling. Large amount of data generated by the study is available through an online tool. The authors have been very responsive to previous reviewers, incorporating additional data to validate the approach among other improvements. There are no outstanding concerns.

We thank the reviewer for appreciating our study and the online resource. We have indeed put considerable effort to make vast amounts of data well accessible to all readers and we are pleased to understand we have succeeded in doing so. We also believe that by addressing the issues raised by the reviewers the manuscript has greatly improved. We are already implementing the method in a larger scale, blinded screen.